# Parametric modeling of under-5 children survival among 30 African countries: Lognormal accelerated failure time gamma shared frailty model

Bikis Liyew[1]*, Kemal Tesfa[2], Kassaye Demeke Altaye[3], Abeje Diress Gelaw[4], Alemu Teshale Bicha[5], Ayanaw Guade Mamo[6], Kassaw Chekole Adane[7]

1 Department of Emergency and Critical Care Nursing, School of Nursing, College of Medicine and Health Sciences, University of Gondar, Gondar, Ethiopia, 2 Department of Internal Medicine, University of Gondar Comprehensive Specialized Hospital, Gondar, Amhara, Ethiopia, 3 Department of Emergency and Critical Care Medicine, School of Medicine, College of Medicine and Health Sciences, University of Gondar, Gondar, Ethiopia, 4 Department of Radiology, School of Medicine, College of Medicine and Health Sciences, University of Gondar, Gondar, Ethiopia, 5 Department of Obstetrics and Gynecology, School of Medicine, College of Medicine and Health Sciences, University of Gondar, Gondar, Ethiopia, 6 Department of Internal Medicine, School of Medicine, College of Medicine and Health Sciences, University of Gondar, Gondar, Ethiopia, 7 Department of Environmental and Occupational Health and Safety, College of Medicine and Health Sciences, Wollo University, Dessie, Ethiopia

* biksliyew16@gmail.com

## Abstract

### Background

Under-five mortality continues to be a serious public health concern in low-and middle-income countries, particularly in Africa. This study investigates the probability of under-five survival and its predictors of mortality in the African continent using a recent demographic health survey from 2014–2022.

### Methods

This study utilized recent Demographic and Health Survey data from 30 African countries, encompassing 226,862 live births. This study employed a multivariable lognormal accelerated failure time gamma shared frailty parametric survival regression analysis to identify the predictors of time-to-death among under-five children in these African nations.

### Result

The overall under-five child mortality rate in Africa was 37.55 per 1,000 live births (95% CI: 37.35, 37.74). In this study, children born in Western Africa; children born in Eastern Africa; children born to mothers aged 15–19 years; maternal educational status; maternal decision autonomy; being female; place of delivery; number of ANC visits; children born among mothers who delivered by Cesarean section; mothers who have multiple birth outcome; children who were second birth order; and third birth order and third birth order; children who were small in size at birth; children who were born from a community with a low women

**Data Availability Statement:** Data is available online, and you can access it from https://www.dhsprogram.com/data/available-datasets.cfm.

**Funding:** The author(s) received no specific funding for this work.

**Competing interests:** The authors declare that they have no conflict of interest.

**Abbreviations:** AHR, Adjusted Hazard Ratio; AIC, Akakie Information Criteria; ANC, Antenatal Care; CI, confidence interval; DHS, Demographic and Health Surveys; KR, Kids Record; LMICs, low- and middle-income countries; SSA, Sub-Saharan Africa.

education; having poor wealth index; respondents working; mothers delivered at the age between 20–35 were significant predictors of survival time to event of under-five children in Africa.

## Conclusion

This study found that the overall under-five mortality rates remain high across Africa. In this study country region, maternal age, maternal education status, maternal age at first birth, respondent's employment status, birth outcome, wealth index, birth order, place of delivery, mode of delivery, women's autonomy in healthcare decision-making, number of antenatal care visits, child's size at birth, sex of the neonate, and community-level women's education were found to be significant predictors of survival time to death of under-five children. Addressing these multilevel factors is crucial for developing targeted interventions to reduce under-five mortality further and improve child survival in African countries.

## 1. Introduction

The under-five mortality rate is the probability of a child dying between the time of delivery and their fifth birthday, expressed per 1000 live births. It is a useful measure of the state of children's health and the general development of nations [1]. Under-five mortality is a key indicator for monitoring progress on the Sustainable Development Goals (SDGs) and Millennium Development Goals (MDGs) [2]. The SDG target is to reduce the under-five mortality rate to less than 25 per 1,000 live births by 2030. However, many countries remain off-track to meeting this ambitious SDG target [3]. The goal of the 2016 Sustainable Development Goals (SDGs) was to substantially reduce the under-five mortality rate to 25 per 1,000 live births or below by 2030 through the execution of various techniques and interventions [4]. Globally, there has been remarkable progress in child survival in the past three decades [5]. Despite remarkable progress, around 5.2 million deaths were from sub-Saharan Africa followed by Central and Southern Asia [5, 6]. The worldwide incidence of under-five fatalities declined from 93 to 41 per 1000 live births between 1990 and 2016 [7]. Under-5 mortality increased substantially, ranging from 25% to 71% in 10 years [8]. Recent estimates from UN IGME showed that the total number of under-5 deaths dropped from 12.6 million in 1990 to 5.4 million in 2017 [9]. In particular, 50% of under-5 deaths occur in sub-Saharan Africa, a region that concentrates 24% of the worldwide under-5 population [10]. In 2018, an estimated 5.3 million under-five deaths occurred worldwide, with 52% of these taking place in sub-Saharan Africa. The average under-five mortality rate in this region was 78 per 1,000 live births, which can be attributed to several contributing factors [7, 11]. Nonetheless, there has been some progress made, as global under-5 mortality rates have declined by half, from 91 deaths per 1,000 live births to 43 deaths per 1,000 live births [12]. From 2000 to 2020, the global under-five mortality rate dropped in half, decreasing from 76 to 37 deaths per 1,000 live births. This significant progress reduced under-five deaths from 12.5 million in 1990 to 5 million in 2020 [3, 13, 14]. The risk of a child dying before turning five is highest in the African Region, which is roughly seven times higher (90 per 1000 live births) compared with the rate in the European Region (12 per 1000 live births) [15].

While many countries have made progress in reducing under-five mortality rates, sub-Saharan Africa still faces high death rates among young children [16]. More recent studies

from Bangladesh, Nigeria, South Sudan, and sub-Saharan Africa demonstrated that characteristics such as residence area [17, 18], family size [19, 20], source of water [20], and toilet facility were determinant factors of under-five mortality rate [21]. Measles, meningitis, and lower respiratory infections are among the vaccine-preventable diseases that have been linked to 20% of under-five child deaths [13]. Place of residence, mothers' education, number of antenatal care visits in pregnancy, and child's sex have all been identified as major predictor variables of under-five child mortality [22–24]. The persistent and substantial disparities in under-five mortality rates across different regions within Africa continue to present a significant, widespread challenge for all countries on the continent. Previous studies failed to account for the survival nature of under-five mortality and neglected the clustering effects in their data, creating a methodological gap. This study addresses these issues by employing a two-level frailty survival analysis to accurately control for cluster effects. Developing a comprehensive understanding and identification of the key determinants associated with the time to death in under-five child mortality, is essential for implementing effective healthcare policies, allocating healthcare resources efficiently, and promoting appropriate interventions to reduce childhood mortality.

## 2. Methods and materials

### 2.1. Study design and area

The data source for this study was the Demographic and Health Surveys (DHS) from 30 African countries, which included Burundi, Ethiopia, Côte d'Ivoire, Egypt, Kenya, Madagascar, Malawi, Mauritania, Namibia, Rwanda, Tanzania, Uganda, Zambia, Zimbabwe, Angola, Cameroon, Chad, the Democratic Republic of the Congo, Gabon, Benin, Burkina Faso, Gambia, Ghana, Guinea, Liberia, Mali, Nigeria, Sierra Leone, Togo, Lesotho, and South Africa. The Demographic and Health Surveys (DHS) are nationally representative surveys that provide valuable data for monitoring various indicators related to population dynamics, nutrition, and health. Permission to use the DHS data for the present study was granted by the Measure DHS program.

### 2.2. Study population and sampling

The target population for this study comprised children under the age of five years residing in the 30 African countries included in the Demographic and Health Surveys (DHS). The specific study population was drawn from the selected enumeration areas (EAs) or clusters within these countries. A multistage cluster sampling technique was employed to recruit the study samples, with the EAs and individual households serving as the primary and secondary sampling units, respectively. It is important to note that the DHS often utilizes an oversampling approach in certain regions or counties, while under-sampling may occur in larger administrative areas. To account for these sampling differences and ensure the representativeness of the data at both the national and sub-national levels, we applied sample weighting as per the DHS recommendations. This statistical adjustment allows for the computation of accurate means, percentages, and regression analyses that are reflective of the true population parameters across the 30 African countries examined in this study. For this study, the mother's most recent births as well as the respondents' regular residences were taken into account, and information was taken from the Kids Record (KR) file. A total of 226, 862 most recent live birth children were taken into account for the analysis (Fig 1).

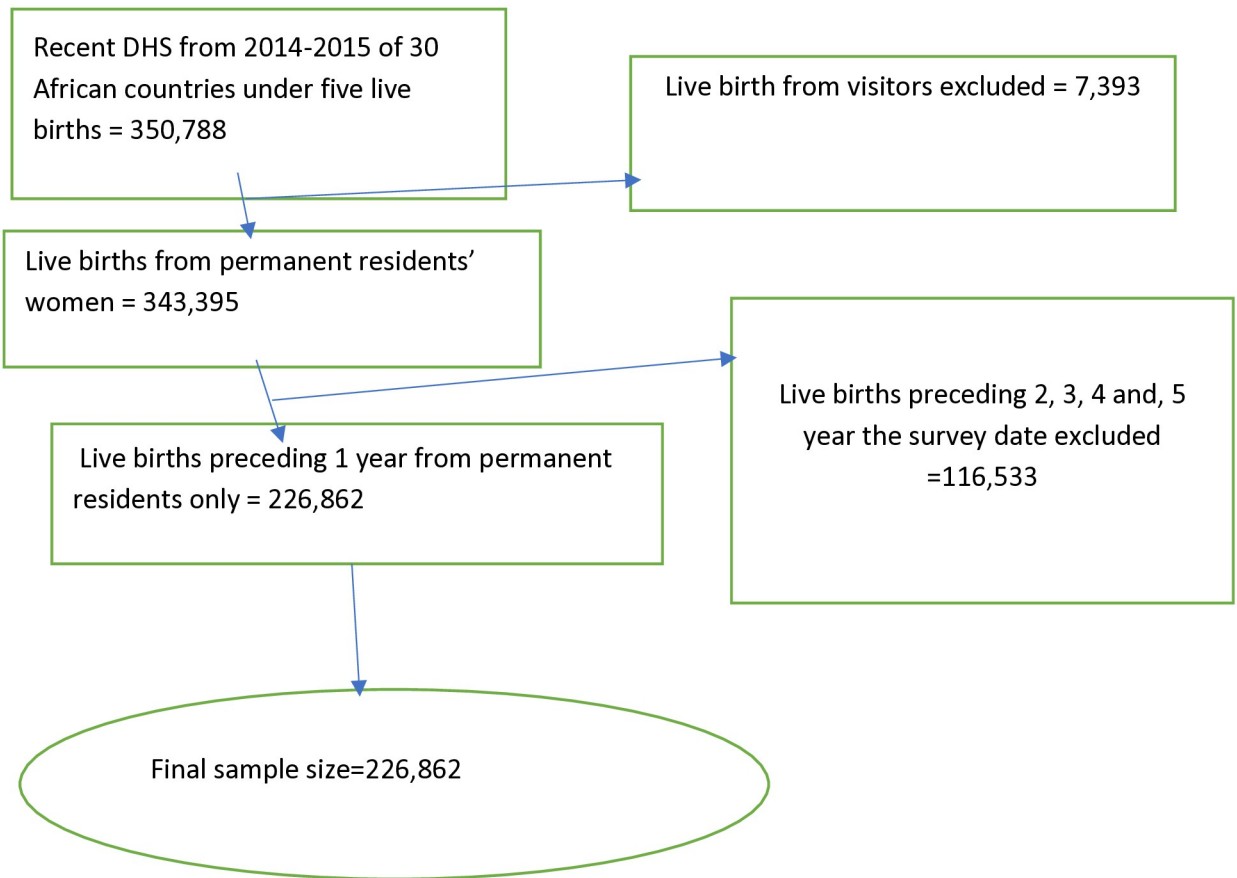

**Fig 1. Sampling procedure to reach the final sample size in 30 African countries from 2014–2022.**

## 2.3. Variables of the study

**2.3.1. Dependent variable.** The time to death of the under-five children before their fifth-year birthday was the dependent variable. The response variable or outcome variables considered in this study are the survival time of a child measured in months from birth until death or censorship of children aged less than 60 months. The outcome variable for this study was under-five children's survival status categorized as being alive (coded as 0) or died (coded as 1).

**2.3.2. Independent variables.** The independent variable considered for this study was at two levels (individual and community). The independent variables considered for this study were categorized as socio-demographic and economic variables. The explanatory variables were categorized into three themes. 1) Demographic variables; residence, marital status, maternal age, and country, 2) socio-economic variables; maternal education, husband education, wealth status, media exposure, and maternal occupation, and 3) maternal obstetric and child-related variables: child age, sex of the child, birth order, birth size, birth outcome, birth size, place of delivery, mode of delivery, women's health care decision making autonomy, number of ANC visits, preceding birth interval, and distance to the health facility. Community-level variables used in the analysis were from two sources; direct community-level variables that were used without any manipulation and aggregated community-level variables that were generated by aggregating individual-level variables at the cluster level. Media exposure was measured by three variables, such as reading the newspaper, listening to the radio,

and watching television. These variables were merged and categorized as no "when there was no exposure to either of the three" and yes "when there was exposure to either reading newspapers, listening to the radio, or watching television". Women's health care decision-making autonomy was assessed in DHS, as a person decides on the respondent's health care. Which was categorized as women participating in making their own health care decisions and didn't participate in making health care decisions (decided by their husband/ partner). Birth weight was categorized as small, average, and large at birth. A small size at birth is defined as a birth weight less than 2500 grams while a birth weight greater than 4000 grams is considered a large size at birth. The variable community development index was created by aggregating individual-level characteristics at the cluster level, and the aggregate variables were categorized. Several households using an improved/unimproved source of drinking water, the number of households using an improved/improved sanitation facility, and the presence of electricity (yes/no) were combined to create a community development index categorized as low, moderate, or good. The other variable wealth index was calculated using household assets (television, bicycle/car, size of agricultural land, number of livestock) and dwelling characteristics (sources of drinking water, sanitation facilities, and materials used to build houses), and the scores were divided into five wealth quintiles (poorest, poorer, medium, richer, and richest).

## 2.4. Data processing and analysis

The variables used in this study were extracted, cleaned, processed, and analyzed by using STATA version 14 software. Descriptive statistics of different variables were presented by text, frequency, cross-tabulation, pie charts, and bar charts. To account for the unequal probability of selection inherent in the DHS sampling design, sample weighting was applied during all statistical analyses. This weighting adjustment was crucial to increasing the representativeness of the survey results and ensuring the findings accurately reflected the target population across the 30 African countries. To determine the significance of bi-variable analysis, the survival analysis model was used. A P-value $\leq 0.25$ in the bi-variable analysis was taken into the multivariable analysis. Kaplan-Meier curves were used to estimate the recurrence-free survival rate, and statistical log-rank and global tests were implemented. Then multivariable lognormal AFT gamma shared frailty parametric survival regression analysis was expressed in the adjusted hazard ratio (HR) with 95% at (P< 0.05) to declare statistically significant predictors of time to death of under-five children. The accelerated failure time frailty model has an unobserved multiplicative effect on the hazard rate for all individuals in the same group. In the shared frailty model, children in the same cluster share the same nuisance (frailty) factor. Parameter $\theta$ provides information on the variability (dependency) of the population in the same cluster. Due to the hierarchical nature of the DHS data, children were nested within EAs; there is a possibility of a clustering effect, which violates the fundamental assumptions of the classical regression model, namely equal variance and independence of observations. We tested for clustering by fitting the frailty model (random effect survival model). If the theta value is significant in the null model (p-value < 0.05), it indicates unobserved heterogeneity or shared frailty. As a result, children in one cluster were more closely related than those in other clusters. Furthermore, the LR-test determines whether the shared frailty model was better fitted to the data than the classical model. Frailty cannot be estimated directly from the data; instead, it is assumed to follow a distribution with a mean of 1 and a variance of 0. Subjects are less likely to be frail if their frailty is less than one and more likely to be frail if their frailty exceeds one. In this study, frailty was modeled using the number of EAs.

## 2.5. Ethics approval and consent to participate

No human participant. Because the study doesn't involve the collection of information from subjects. Consent to participate is not applicable. Since the study is a secondary data analysis based on demographic health survey data. Therefore, more details regarding DHS data and ethical standards are available online at https://dhsprogram.com/data/available-datasets.cfm.

## 3. Results

### 3.1. Descriptive characteristics of the study participants

A total of 226,862 unweighted live births were included for analysis in the one year preceding each country's interview or survey of live birth children from permanent residents' mothers, excluding visitors. About 164,680.04 (73.11%) of the children were born to mothers aged 20–35 years, and more than half of the respondents, 148,688.02 (66.01%) were rural residents. About 95,401.4849 (42.35%) and 84,310.836 (37.43%) of the children belonged to poor and rich households, respectively. Nearly one-third (77,230.449, (34.29%) of the children were born to mothers who didn't have formal education, and the majority (144,530.67, (64.25%) of the respondents had media exposure. Regarding birth size, about 20,708.933 (9.19%) of them were small at birth. Out of the total recent live births under five children, nearly half (114,856.73) (50.99%) are male. Out of the total recent live births under five children, nearly half (114,856.73) (50.99%) are male. About 143,803.264 (63.84%) of the communities had a high proportion of community ANC utilization (Table 1).

### 3.2. Under-five children mortality rates in Africa

Out of 226,862 live births, 8,971 under-five children died. Overall, the under-five child mortality rate in Africa was 37.55% with a 95% confidence interval: of 37.36–37.74) per 1000 live births. The under-five mortality rate varies by country, which ranges from 13.25 per 1000 live births in Egypt to 62.71 per 1000 live births in Nigeria (Table 2 and Fig 2).

 **3.2.1. Subgroup analysis by country region.** The present study revealed that the highest under-five mortality rate in African country regions in Western Africa was 48.71% per 1000 live births, and the lowest under-five mortality rate seen in Northern Africa was 17.72 per 1000 live births (Fig 3).

### 3.3. Incidence rate of under-five mortality in Africa

A total of 226,862 live births of under-five children were followed for 5 years with 199503071 person days' observations. The overall mean follow-up time was 879.4028 days (95% CI: 876.82, 881.98). From the total live births under 5 children included in the analysis, about 8,971 (3.95%) (95%CI: 3.875, 4.0354) died in the entire follow-up period. The overall incidence rate of under-five child mortality in the cohort was 0.045 per 1000-person day's observations (95%CI, 0.044, 0.0459) (Table 3).

### 3.4. Survival and failure function pattern of Under-five children

The cumulative probabilities of death-free survival at 1, 7, 14, and 28 days were 0.9894, 0.9836, 0.9822, and 0.9811, respectively. The probability of survival without death or event decreases as the number of follow-up days increases, with the lowest survival probability (95.99%) occurring in the final days of observation (Fig 4 and S1 Table). The cumulative probabilities of an event (death) at 1, 7, 14, and 28 days were 0.0106, 0.0164, 0.0178, and 0.0189 (Fig 5 and S2 Table).

**Table 1. Summary results of covariates of time-to-death for under-five children in Africa using recent DHS, 2014–2022.**

| Respondent's characteristic | Categories | Weighted Frequency | Percentage (%) |
|---|---|---|---|
| Place of residence | Urban | 76,567.037 | 33.99 |
| | Rural | 148,688.02 | 66.01 |
| Maternal age | 15–19 | 16,277.702 | 7.23 |
| | 20–35 | 164,680.04 | 73.11 |
| | 36–49 | 44,297.317 | 19.67 |
| Maternal Education status | No education | 77,230.449 | 34.29 |
| | Primary education | 71,861.1472 | 31.90 |
| | Secondary education | 64,817.882 | 28.78 |
| | Higher education | 11,345.581 | 5.04 |
| Maternal age at first birth | 8–19 | 128,362.06 | 56.99 |
| | 20–35 | 96,280.766 | 42.74 |
| | 36–49 | 612.231957 | 0.27 |
| wealth index combined | Poor | 95,401.4849 | 42.35 |
| | Middle | 45,542.739 | 20.22 |
| | Rich | 84,310.836 | 37.43 |
| Sex of household head | Male | 179,123.83 | 79.52 |
| | Female | 46,131.228 | 20.48 |
| Respondent working | No | 70,025.135 | 31.09 |
| | Yes | 155,229.92 | 68.91 |
| Birth outcome | Single | 220,829.97 | 98.04 |
| | Multiple | 4,425.0919 | 1.96 |
| Birth order | First | 47,945.065 | 21.28 |
| | Second | 44,333.266 | 19.68 |
| | Third | 37,493.367 | 16.64 |
| | 4+ | 95,483.362 | 42.39 |
| Place of delivery | Home | 63,274.244 | 28.09 |
| | Health facility | 161,980.82 | 71.91 |
| Mode of delivery | Vaginal | 205,773.454 | 91.35 |
| | C/S | 19,481.606 | 8.65 |
| Women's healthcare decision making autonomy | Respondent alone | 32,592.913 | 14.47 |
| | Jointly with their husband/parent | 79,947.532 | 35.49 |
| | Husband/parent alone | 112,714.62 | 50.04 |
| Number of ANC visit | No visits | 27,996.216 | 12.43 |
| | 1–3 | 65,956.53 | 29.28 |
| | ≥4 | 131,302.31 | 58.29 |
| Children's s' size at birth | Small | 20,708.933 | 9.19 |
| | Average | 133,730.62 | 59.37 |
| | Large | 70,815.503 | 31.44 |
| Sex of children | Male | 114,856.73 | 50.99 |
| | Female | 110,398.33 | 49.01 |
| Household development index | Low | 17,228.693 | 7.65 |
| | Moderate | 63,819.39 | 28.33 |
| | Good | 144,206.98 | 64.02 |
| Household media exposure | No | 80,428.269 | 35.75 |
| | Yes | 144,530.67 | 64.25 |
| Community Media exposure | Low | 106,395.25 | 47.23 |
| | High | 118,859.81 | 52.77 |

(*Continued*)

**Table 1.** (Continued)

| Respondent's characteristic | Categories | Weighted Frequency | Percentage (%) |
|---|---|---|---|
| Community-women education | Low | 127,888.64 | 56.78 |
| | High | 97,366.42 | 43.22 |
| Community poverty | Low | 115,533.96 | 51.29 |
| | High | 109,721.1 | 48.71 |
| Community ANC utilization | Low | 81,451.796 | 36.16 |
| | High | 143,803.264 | 63.84 |
| Community development index | Low | 107,049.89 | 47.52 |
| | High | 118,205.17 | 52.48 |

**Table 2. Under-five children mortality rates in Africa using recent standard DHS, from 2014–2022.**

| Countries | Weighted censored | Weighted died | Total weighted frequency | U5MR/1000 live births) | 95% CI LB | 95% CI UB |
|---|---|---|---|---|---|---|
| Angola | 8,126.35 | 277.39517 | 8,403.74 | 32.89 | 29.37 | 36.82 |
| Burkina Faso | 6,128.77 | 153.0552 | 6,281.83 | 26.03 | 22.38 | 30.245 |
| Benin | 8,441.45 | 401.35515 | 8,842.80 | 43.67 | 39.59 | 48.14 |
| Burundi | 8,577.93 | 317.0199 | 8,894.95 | 32.85 | 29.29 | 36.83 |
| DR Congo | 10,386.61 | 519.71702 | 10,906.33 | 51.44 | 47.48 | 55.69 |
| Cote devour | 4,930.46 | 232.41279 | 5,162.87 | 45.41 | 40.229 | 51.23 |
| Cameron | 6,026.32 | 265.07866 | 6,291.40 | 46.46 | 41.45 | 52.04 |
| Egypt | 10,762.27 | 143.01504 | 10,905.28 | 13.25 | 11.26 | 15.58 |
| Ethiopia | 7,152.35 | 301.71985 | 7,454.07 | 38.54 | 34.29 | 43.279 |
| Gabon | 4,148.08 | 102.681987 | 4,250.77 | 29.73 | 25.05 | 35.24 |
| Ghana | 3,934.82 | 107.588237 | 4,042.41 | 28.55 | 23.92 | 34.04 |
| Gambia | 4,983.59 | 161.91715 | 5,145.51 | 35.93 | 31.34 | 41.15 |
| Guiana | 5,079.82 | 332.070993 | 5,411.89 | 60.38 | 54.35 | 67.02 |
| Kenya | 8,950.32 | 205.75389 | 9,156.08 | 24.08 | 21.26 | 27.26 |
| Liberia | 3,716.42 | 196.02134 | 3,912.44 | 49.99 | 43.77 | 57.04 |
| Lesotho | 2,248.39 | 127.40623 | 2,375.80 | 53.14 | 44.83 | 62.88 |
| Madagascar | 8,777.95 | 351.059468 | 9,129.01 | 36.79 | 33.14 | 40.84 |
| Mali | 6,294.41 | 289.84047 | 6,584.25 | 43.31 | 38.56 | 48.61 |
| Mauritania | 7,292.18 | 210.95805 | 7,503.14 | 29.57 | 25.95 | 33.66 |
| Malawi | 12,981.59 | 420.69915 | 13,402.29 | 30.73 | 27.93 | 33.79 |
| Nigeria | 20,320.52 | 1,338.90 | 21,659.41 | 62.71 | 59.54 | 66.03 |
| Rwanda | 6,060.34 | 140.31049 | 6,200.65 | 23.05 | 19.56 | 27.14 |
| Seralione | 6,774.26 | 495.49484 | 7,269.76 | 64.11 | 58.72 | 69.96 |
| Chad | 10,341.55 | 666.70426 | 11,008.26 | 57.35 | 53.149 | 61.85 |
| Togo | 4,559.95 | 186.36933 | 4,746.32 | 39.13 | 34.05 | 44.93 |
| Tanzania | 5,529.10 | 151.2936 | 5,680.39 | 25.31 | 21.51 | 29.76 |
| Uganda | 9,457.33 | 347.56865 | 9,804.90 | 36.96 | 33.42 | 40.85 |
| South Africa | 2,844.29 | 84.435019 | 2,928.72 | 28.51 | 23.05 | 35.22 |
| Zambia | 6,912.95 | 215.35224 | 7,128.30 | 30.31 | 26.59 | 34.53 |
| Zimbabwe | 4,625.22 | 146.28277 | 4,771.50 | 29.45 | 24.95 | 34.74 |

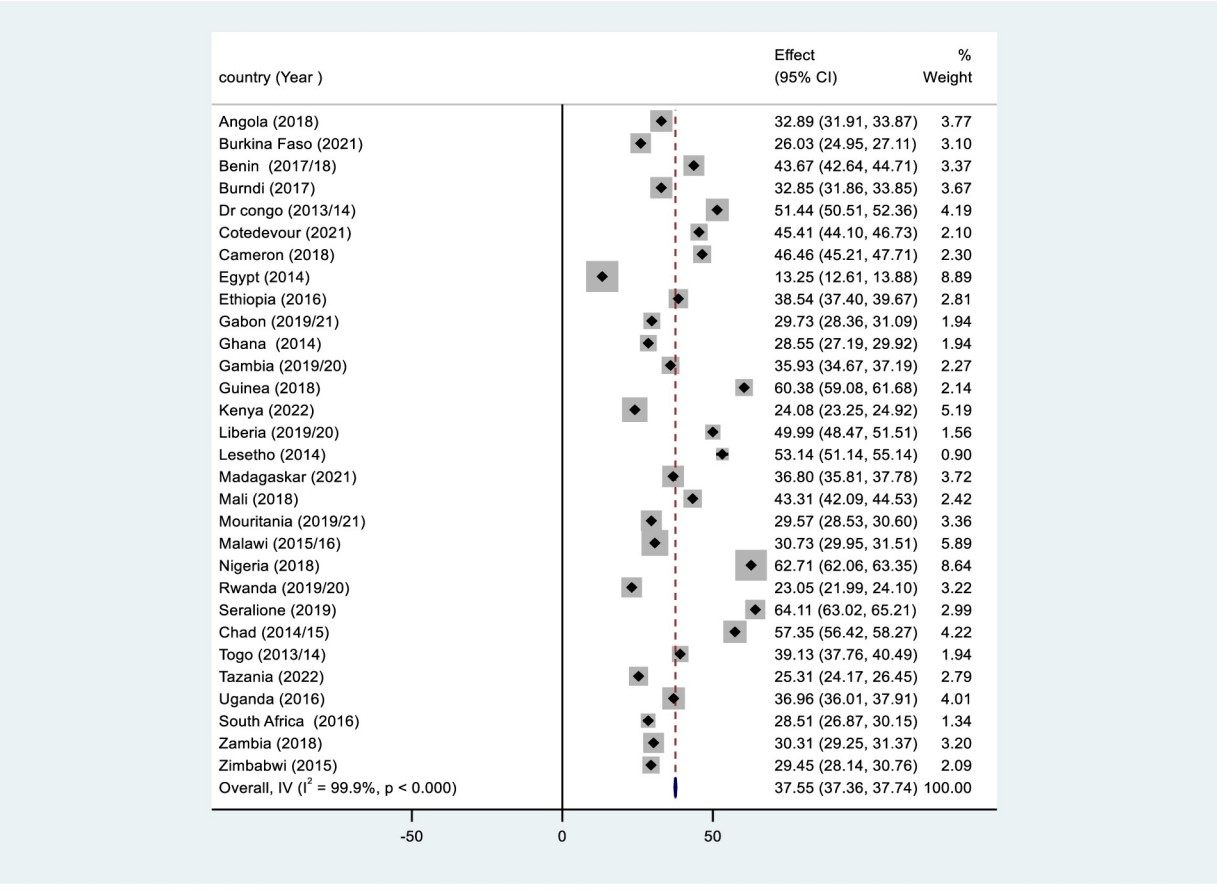

**Fig 2. Under-five children mortality rates in Africa using recent DHS data from 2014–2024.**

## 3.5. Survival functions of different categorical predictor variables

In this study, we released that maternal birth outcome status was a strong predictor of mortality. Those with single birth status had a higher event-free survival rate at the end of follow-up than those with multiple birth outcomes. When one survivorship function is placed ahead of another, the group indicated by the upper curve has a higher survival rate. The survival plot of time-to-survey time-to-death of under-5 children by birth outcome plot demonstrates that there is a difference in risk between single and multiple birth outcomes among mothers (Fig 6). The risk of death in twin children is higher than in singletons, and also there is a difference in risk between groups of family size, and place of delivery (Figs 7 and 8).

**3.5.1. Log rank tests.** In this study, the log-rank test was used at a 5% level of significance to validate the differences in the survival time of each factor. The null hypothesis, that there is no change in the likelihood of an event occurring at any time point, was tested through a log-rank test. This test shows that current maternal age, maternal age at first childbirth, maternal education, wealth status, sex of the child, children's size at birth, mode of delivery, birth order, birth outcome, number of ANC visits, women's health care decision-making autonomy, respondents' working, place of residence, country region, media exposure, community women's education, community poverty, and community ANC utilization had a statistically significant difference in experiencing death events, whereas household media exposure, sex of the

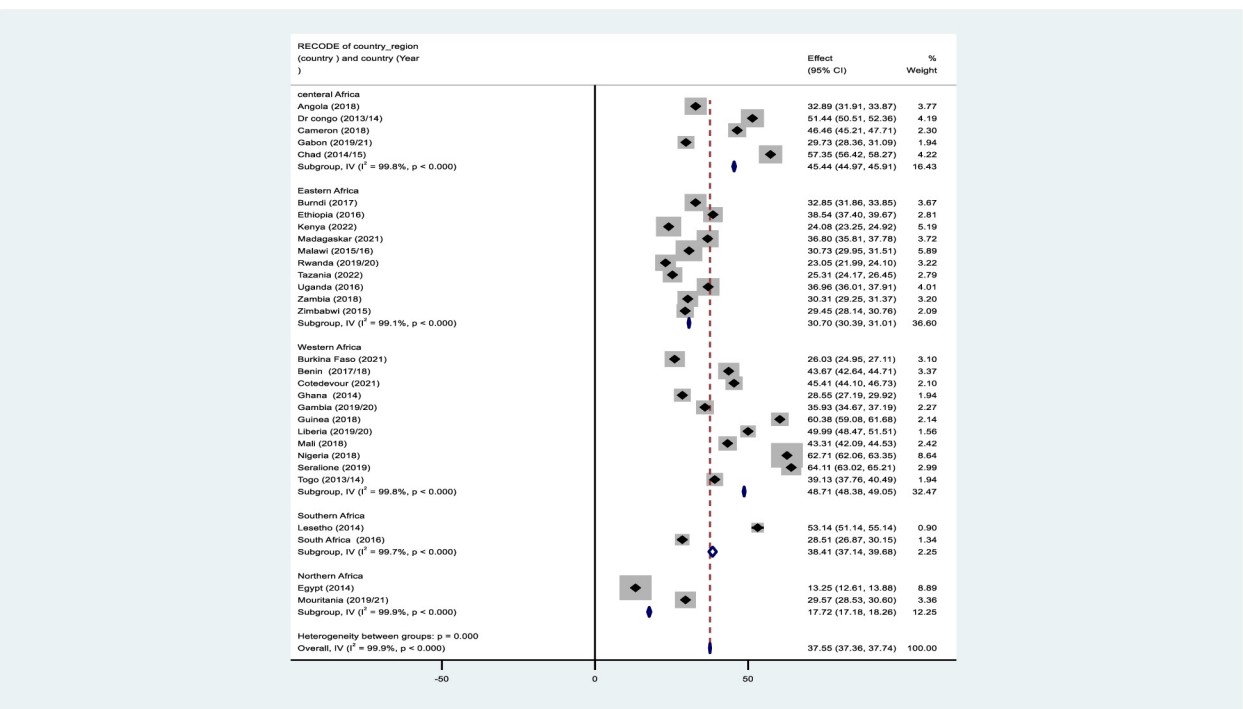

**Fig 3. The Under-five children mortality rate per 1000 live births in African countries by region using recent DHS from 2014–2022.**

household head, and mode of delivery did not statistically significant in experiencing death events (Table 4).

### 3.6. Test of proportional-hazards assumption

The global Schoenfeld residuals test (global and scaled) was used to test the proportional hazard assumption for all potential under five mortality predictors. The global Schoenfeld residual test resulted in a p-value < 0.05. Therefore, the temporal distribution in our data did not meet the assumption of proportional hazard, because the global Schoenfeld residuals test used to check the proportional hazard assumption revealed that the proportional hazard assumption was violated, indicating that the Cox-proportional model is not appropriate. Therefore, we move on to the parametric survival models (Table 5).

### 3.7. Modeling of parametric AFT shared frailty survival analysis

Since the proportional hazards assumption was not satisfied, the accelerated failure time model is also an alternative model for the analysis of this data. Therefore, the global null hypothesis that the proportionality assumption holds is rejected, and the PH model is not appropriate here. In the following, we proceed to model the attrition lifetime data using the AFT model. We fitted the data using an accelerated failure time model with exponential, Weibull, lognormal, and log-logistic as a baseline distribution with different frailty distributions. The frailty model has an unobserved multiplicative effect on the hazard rate for all individuals in the same group. In the shared frailty model, children in the same cluster share the same nuisance (frailty) factor. Due to the hierarchical nature of the DHS data, children were nested within enumeration areas. There is a possibility of a clustering effect, which violates the

**Table 3. The incidence rate per 1000 live births of under-five children in African countries using recent DHS from 2014–2022.**

| Country | Year | Person time | Failure (died) | Incidence rate per 1000-person day's | 95% CI /1000-person days |
|---|---|---|---|---|---|
| Burundi | 2017 | 5807815 | 283 | 0.04873 | 0.0434, 0.0547 |
| Ethiopia | 2016 | 4936874 | 273 | 0.0553 | 0.0491, 0.0623 |
| Cote devour | 2021 | 2769636 | 251 | 0.09063 | 0.0801,0.1026 |
| Egypt | 2014 | 19311544 | 144 | 0.007457 | 0.00633, 0.00878 |
| Kenya | 2022 | 5236632 | 243 | 0.0464 | 0.0409, 0.0526 |
| Madagascar | 2021 | 6820874 | 339 | 0.0497 | 0.0447, 0.0553 |
| Malawi | 2015/16 | 10223538 | 410 | 0.0401 | 0.0364, 0.0442 |
| Mauritania | 2019/21 | 5078021 | 220 | 0.04332 | 0.038, 0.0494 |
| Rwanda | 2019/20 | 4657994 | 140 | 0.03006 | 0.0255, 0.0355 |
| Tanzania | 2022 | 2914557 | 142 | 0.04872 | 0.0413, 0.0574 |
| Uganda | 2016 | 6844444 | 367 | 0.05362 | 0.0484, 0.0594 |
| Zambia | 2018 | 5300785 | 218 | 0.04113 | 0.036, 0.047 |
| Zimbabwe | 2015 | 3596627 | 136 | 0.03781 | 0.032, 0.0447 |
| Angola | 2015/16 | 5633085 | 291 | 0.05166 | 0.0461, 0.0579 |
| Cameroon | 2018 | 4113701 | 283 | 0.06879 | 0.0612, 0.0773 |
| Chad | 2014/15 | 18662711 | 630 | 0.03376 | 0.0312, 0.0365 |
| DR Congo | 2013/14 | 19037282 | 573 | 0.0301 | 0.0277, 0.0327 |
| Gabon | 2019/21 | 3066359 | 128 | 0.04174 | 0.0351, 0.0496 |
| Benin | 2017/18 | 5863911 | 385 | 0.06566 | 0.0594, 0.0726 |
| Burkina Faso | 2021 | 3385503 | 165 | 0.04874 | 0.0418, 0.0568 |
| Gambia | 2019/20 | 3814573 | 200 | 0.05243 | 0.0456, 0.0602 |
| Ghana | 2014 | 7351879 | 120 | 0.01632 | 0.0136, 0.0195 |
| Guinea | 2018 | 3711419 | 329 | 0.08865 | 0.0796, 0.0988 |
| Liberia | 2019/20 | 3035855 | 208 | 0.06851 | 0.0598, .0.0785 |
| Mali | 2018 | 4115336 | 274 | 0.06658 | 0.0591, 0.0749 |
| Nigeria | 2018 | 14107869 | 1348 | 0.09555 | 0.0906, 0.1008 |
| Sierra Leone | 2019 | 5091578 | 469 | 0.09211 | 0.0841, 0.1008 |
| Togo | 2014 | 8491784 | 192 | 0.02261 | 0.0196, 0.026 |
| Lesotho | 2014 | 4076140 | 127 | 0.03116 | 0.0262, 0.0371 |
| South Africa | 2016 | 2444745 | 83 | 0.03395 | 0.0274, 0.0421 |
| Total | | 1.995e+08 | 8971 | 0.04497 | 0.044, 0.0459 |

fundamental assumption of the classical regression model, namely equal variance and independence of observations. We tested for clustering by fitting the AFT frailty model (random effect survival model). If the theta value is significant in the null model (p-value < 0.05), it indicates unobserved heterogeneity or shared frailty. As a result, children in one cluster were more closely related than those in other clusters. Furthermore, the LR-test determines whether the AFT shared frailty model was better fitted to the data than the classical model. Frailty cannot be estimated directly from the data; instead, it is assumed to follow a distribution with a mean 1 and a variance 0. Subjects are less likely to be frail if their frailty is less than one and more likely to be frail if their frailty exceeds one. In this study, frailty was modeled using the number of EAs. For instance, exponential ($\theta = 0.0593029$), LR test of theta = 0: chibar2 (01) = 79.15, Prob > = chibar2 < 0.001). The Weibull, exponential, log-normal, and log-logistic baseline AFT shared frailty models all showed significant variance in the random effect ($\theta$). The highest value was obtained with the gamma distribution ($\theta = 0.05993$), followed by the inverse Gaussian distribution ($\theta = 0.0579$) in the exponential baseline hazard function. In all baselines

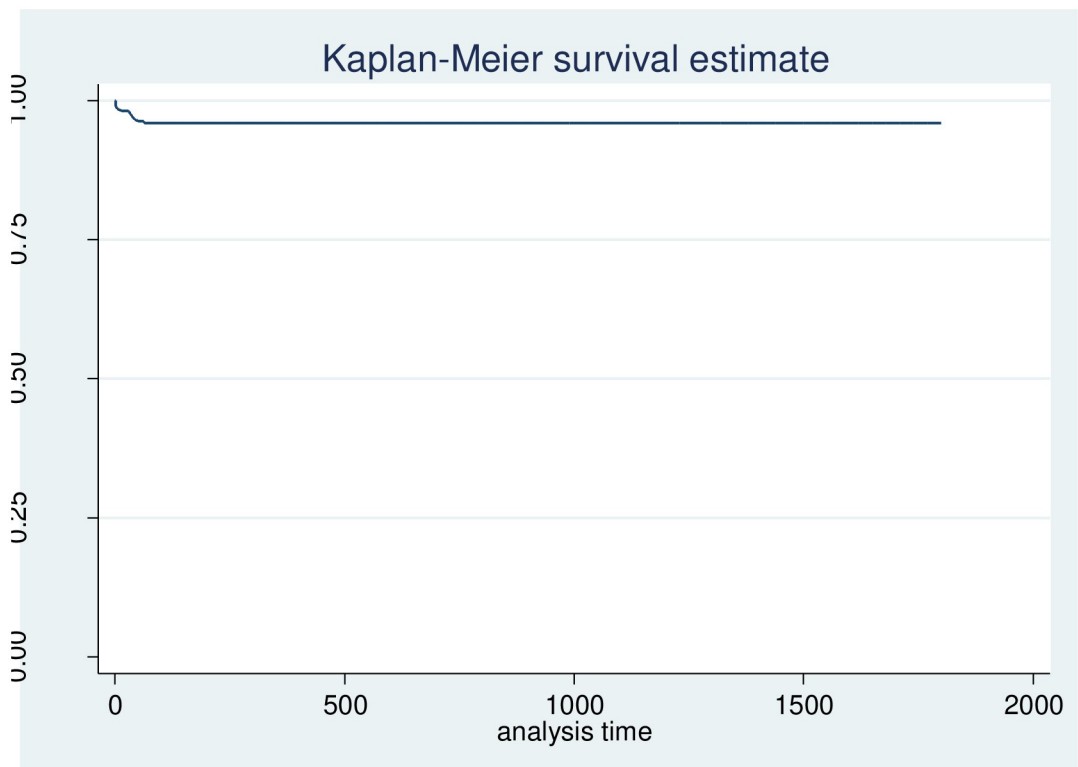

**Fig 4. Overall Kaplan-Meier estimation of the survival function of follow-up children in Africa using recent DHS 2023 (n = 226862).**

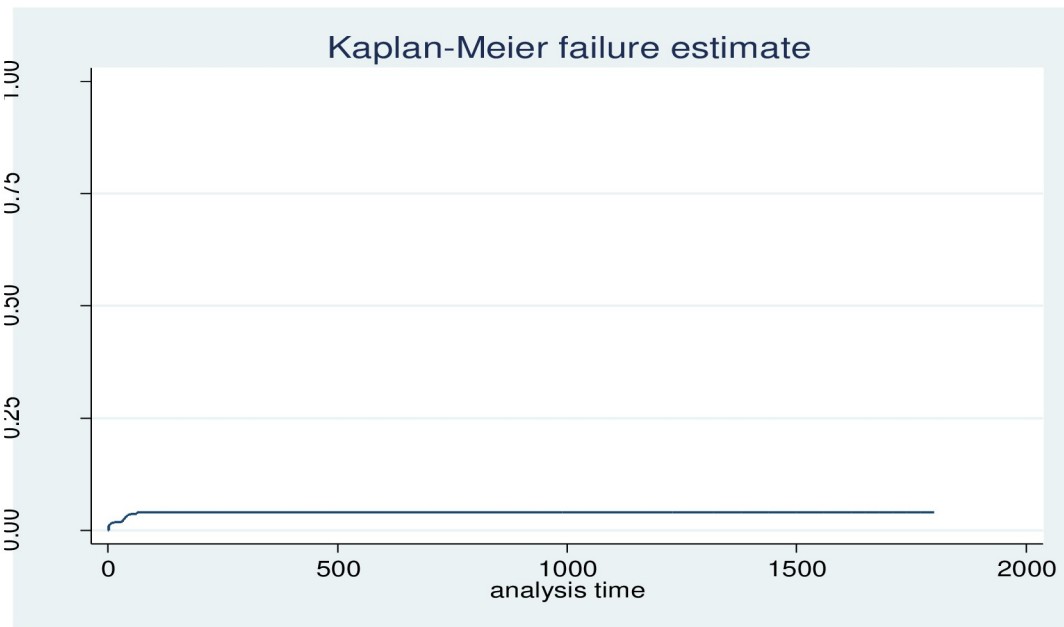

**Fig 5. Overall Kaplan-Meier estimation of failure function follow-up children.**

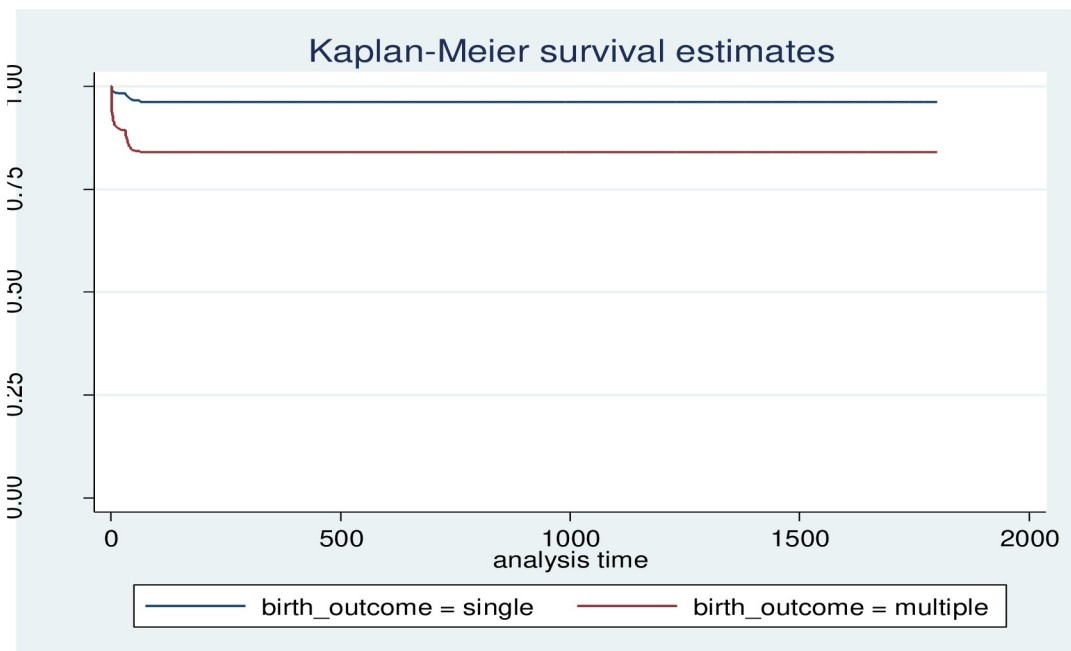

**Fig 6. Survival time-to-death of under 5 children by birth outcome.**

AFT shared frailty model suggests that there is unobserved heterogeneity or shared frailty. Furthermore, because the LR-test was significant, the shared frailty model provided the best fit for the data. As a result, children in one cluster had a closer relationship than those in other groups (Table 6).

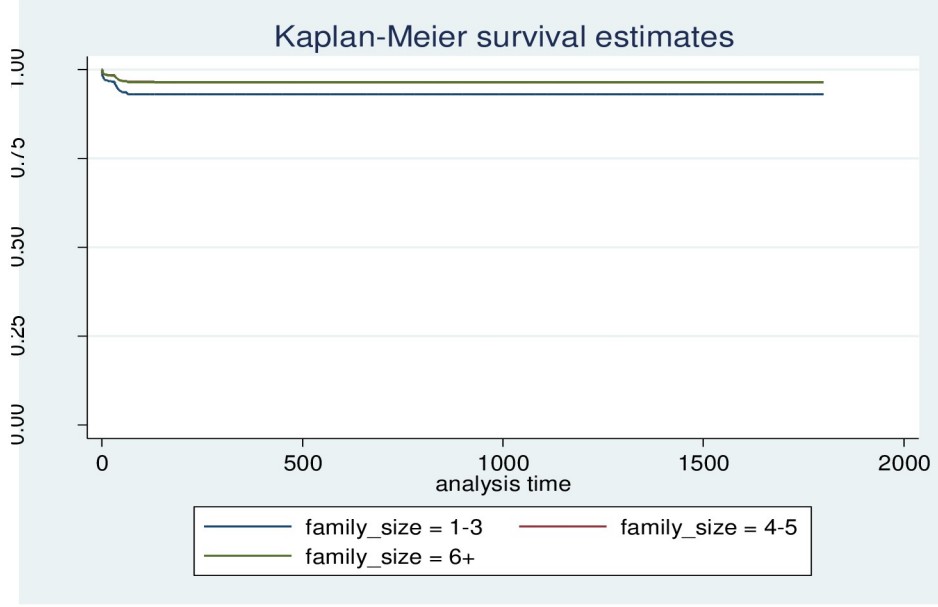

**Fig 7. Survival time-to-death of under 5 children by family size.**

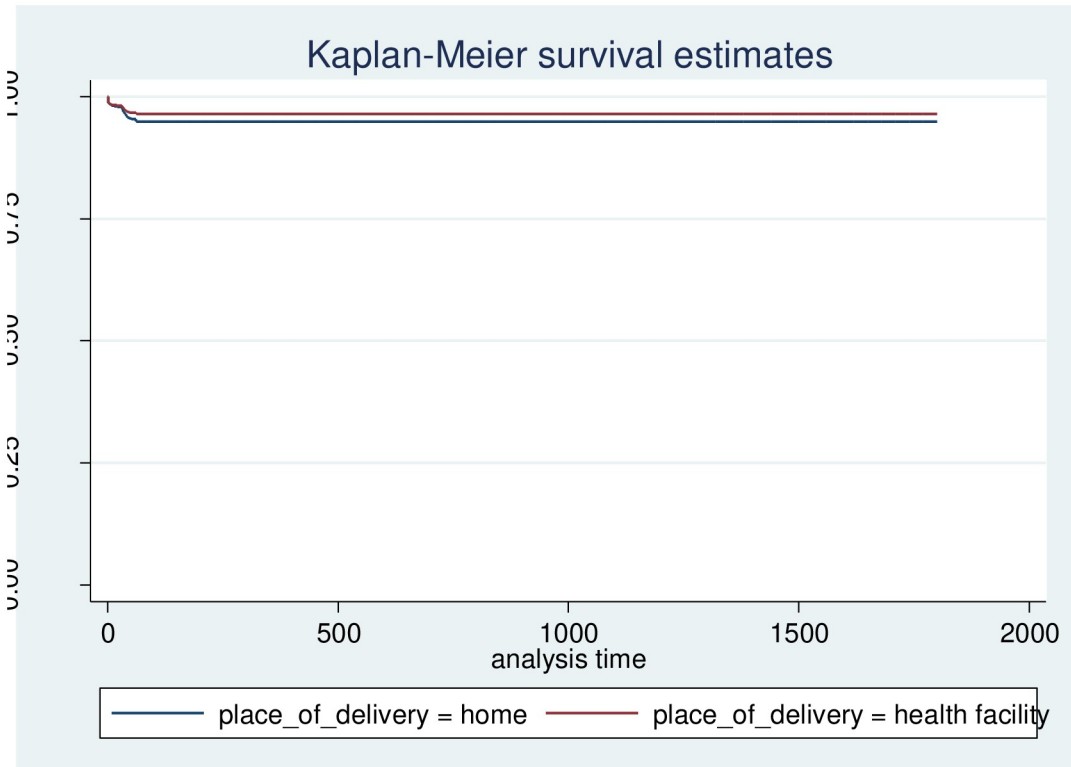

**Fig 8. Survival time-to-death of under 5 children by place of delivery.**

## 3.8. Model comparison

There are numerous procedures for selecting a model from a set of models. There are several methods for model selection. The final model's goodness of fit among parametric shared frailty models was compared using the log-likelihood ratio, deviance, Akaike's information criterion (AIC), and the Bayesian information criterion (BIC). Based on this information, the final fitted analysis model, the lognormal AFT gamma shared frailty model, was found to be the best fit of the five parametric models (AIC = 113660.7) (Table 7).

## 3.9. Predictors of under 5 mortalities in Africa

The effect of a covariate is to accelerate or decelerate the attrition lifetime. To understand this better, a time ratio (TR), also called the acceleration factor, is estimated. The acceleration factors for a given covariate are the exponents of the corresponding coefficient. A positive coefficient means the effect of the covariate is to prolong the survival time, while a negative coefficient is to shorten the time to attrition. The current study's survival time to event was significantly related to country region, maternal age, maternal education status, maternal age at first birth, respondents working, birth outcome, wealth index combined, birth order, place of delivery, mode of delivery, women's health care decision making autonomy, number of ANC visits, child size at birth, sex of neonates, and community women education. This indicates that they are important factors for the survival time to death of under-five children. An acceleration factor or time ratio greater than 1 specifies prolonging the survival time of the event, and an acceleration factor less than 1 indicates shortening the survival time of the event.

**Table 4. Log-rank test of survival time among the different groups of covariates for children in Africa using recent DHS 2023 (n = 226,862).**

| Respondent's characteristic | Categories | Status of event | | Log-rank test | P value |
|---|---|---|---|---|---|
| | | Event observed | Event expected | | |
| Place of residence | Urban | 2581 | 2956.41 | 71.37 | P<0.001 |
| | Rural | 6390 | 6014.59 | | |
| Country | Central Africa | 1905 | 1641.83 | 575.13 | P<0.001 |
| | Eastern Africa | 2551 | 3234.77 | | |
| | Western Africa | 3941 | 3149.08 | | P<0.001 |
| | Northern Africa | 364 | 734.79 | | |
| | Southern Africa | 210 | 210.54 | | |
| Maternal age | 15–19 | 887 | 651.95 | 449.17 | P<0.001 |
| | 20–35 | 5638 | 6529.70 | | |
| | 36–49 | 2446 | 1789.36 | | |
| Maternal Education status | No education | 3863 | 3196.64 | | |
| | Primary education | 2841 | 2852.77 | 307.26 | P<0.001 |
| | Secondary education | 2035 | 2513.26 | | |
| | Higher education | 232 | 408.33 | | |
| Maternal age at first birth | 8–19 | 5575 | 5136.08 | 91.89 | P<0.001 |
| | 20–35 | 3365 | 3810.71 | | |
| | 36–49 | 31 | 24.20 | | |
| wealth index combined | Poor | 4491 | 4046.20 | | |
| | Middle | 1850 | 1805.41 | 127.25 | P<0.001 |
| | Rich | 2630 | 3119.39 | | |
| Sex of household head | Male | 7073 | 7083.43 | 0.07 | 0.7866 |
| | Female | 1898 | 1887.57 | | |
| Respondents working | No | 2311 | 2872.47 | 162.04 | P<0.001 |
| | Yes | 6660 | 6098.53 | | |
| Birth outcome | Single | 8269 | 8808.89 | 1838.52 | P<0.001 |
| | Multiple | 702 | 162.11 | | |
| Birth order | First | 1843 | 1876.59 | 325.44 | P<0.001 |
| | Second | 1357 | 1740.13 | | |
| | Third | 1132 | 1487.60 | | |
| | 4+ | 4639 | 3866.68 | | |
| Child size at birth | Small | 1456 | 852.96 | 494.40 | P<0.001 |
| | Average | 4737 | 5330.84 | | |
| | Large | 2778 | 2787.20 | | |
| Place of delivery | Home | 3355 | 2615.28 | 296.42 | P<0.001 |
| | Health facility | 5616 | 6355.72 | | |
| Mode of delivery | Vaginal | 8235 | 8227.45 | 0.08 | 0.7721 |
| | C/S | 736 | 743.55 | | |
| Women's healthcare decision making autonomy | Respondent alone | 1084 | 1295.59 | 190.45 | |
| | Jointly with their husband/par | 2755 | 3194.70 | | P<0.001 |
| | husband/partner alone | 5132 | 4480.71 | | |
| Number of ANC visit | No visits | 1789 | 1150.42 | 433.14 | P<0.001 |
| | 1–3 visits | 2637 | 2644.11 | | |
| | > = 4 visits | 4545 | 5176.47 | | |
| Sex of child | Male | 4885 | 4558.97 | 47.59 | P<0.001 |
| | Female | 4086 | 4412.03 | | |

(*Continued*)

**Table 4.** (Continued)

| Respondent's characteristic | Categories | Status of event | | Log-rank test | P value |
| --- | --- | --- | --- | --- | --- |
| | | Event observed | Event expected | | |
| Media exposure | No | 3749 | 3340.69 | 79.92 | 0.2006 |
| | Yes | 5201 | 5609.31 | | |
| Community-women education | Low | 5888 | 5259.02 | 182.49 | P<0.001 |
| | High | 3083 | 3711.98 | | |
| Community poverty | Low | 4143 | 4423.02 | 35.10 | 0.0039 |
| | High | 4828 | 4547.98 | | |
| Community ANC utilization | Low | 3819 | 3366.71 | 97.63 | P<0.001 |
| | High | 5152 | 5604.29 | | |

The survival time to death of children born in Western Africa was shortened by a factor of 0.494 ($\Phi$ = 0.494, 95% CI: (0.268, 0.909)) by keeping the remaining covariates constant. The survival time to death of children born in Eastern Africa was increased by a factor of 2.855 ($\Phi$ = 2.855, 95% CI: 1.555, 5.243) by keeping the remaining covariates constant. The estimated acceleration factor of children born to mothers aged 15–19 years was estimated to be 5.366, indicating that the survival time to death of children born to mothers aged 15–19years was increased by 5.366 ($\Phi$ = 5.366, 95% CI: 3.668, 7.848) by keeping the remaining covariates constant as compared with children born to a mother aged between 36–49 years. The estimated

**Table 5. Schoenfeld residual test for checking proportional hazard assumptions for the incidence of under-five child mortality and its predictors among live births in the 30 African countries.**

| Variables | Rho | chi2 | Df | Chi2 Prob >chi2 |
| --- | --- | --- | --- | --- |
| Place of residence | 0.00321 | 0.09 | 1 | 0.7613 |
| Place of delivery | -0.07718 | 54.60 | 1 | P<0.001 |
| Mode of delivery | -0.03605 | 12.51 | 1 | 0.0004 |
| Maternal age at first birth | -0.01986 | 3.51 | 1 | 0.0611 |
| Maternal age | 0.02615 | 6.84 | 1 | 0.0089 |
| Maternal Education | -0.02564 | 6.07 | 1 | 0.0138 |
| Respondents working | 0.03251 | 9.59 | 1 | 0.0020 |
| wealth index combined | -0.01578 | 2.24 | 1 | 0.1347 |
| Birth outcome | -0.08483 | 65.47 | 1 | P<0.001 |
| Children's size at birth | 0.10175 | 112.05 | 1 | P<0.001 |
| Birth order | -0.00429 | 0.17 | 1 | 0.6844 |
| ANC visits | 0.02151 | 4.39 | 1 | 0.0362 |
| Country Region | -0.01548 | 2.21 | 1 | 0.1369 |
| Sex o children | 0.05727 | 29.46 | 1 | P<0.001 |
| Sex of household head | 0.00917 | 0.75 | 1 | 0.3862 |
| Media exposure | -0.02210 | 4.52 | 1 | 0.0334 |
| Women's healthcare decision making autonomy | 0.00508 | 0.24 | 1 | 0.6252 |
| Community-women education | -0.01184 | 1.26 | 1 | 0.2616 |
| Community poverty level | -0.00696 | 0.43 | 1 | 0.5101 |
| Community ANC utilization | -0.02272 | 4.63 | 1 | 0.0314 |
| Global test | | 492.36 | 20 | P<0.001 |

**Significant at p-value p<0.001.

**Table 6. Variance of the random effect (θ) for null model of parametric survival AFT shared frailty baseline hazard functions.**

| Model Distribution | Frailty | Theta | LR test of theta | p-value |
|---|---|---|---|---|
| Exponential | Gamma | 0.0593029 | 79.15 | P<0.001 |
| | Inverse Gaussian | 0.0579301 | 78.52 | P<0.001 |
| Weibull | Gamma | 0.0297125 | 30.82 | P<0.001 |
| | Inverse Gaussian | 0.0293251 | 30.61 | P<0.001 |
| Log-normal | Gamma | 0.0280864 | 28.65 | P<0.001 |
| | Inverse Gaussian | 0.0279112 | 28.50 | P<0.001 |
| Log–logistic | Gamma | 0.0290225 | 29.87 | P<0.001 |
| | Inverse Gaussian | 0.0287138 | 29.68 | P<0.001 |

acceleration factor for mothers who didn't attend formal education was estimated to be 0.286, indicating that the survival time to death of mothers who had no formal education was decreased by a factor of 0.286 (Φ = 0.286, 95% CI: (0.16, 0.512) than mothers who had higher educational status by keeping the remaining covariates constant. The estimated acceleration factor for mothers who had primary education was estimated to be 0.215, indicating that the survival time to death of mothers who had primary education was decreased by a factor of 0.215 (Φ = 0.215, 95% CI: 0.122, 0.379) as compared to mothers who had higher educational status by keeping the remaining covariates constant. The estimated acceleration factor for mothers who attend secondary education was estimated to be 0.351, indicating that the survival time to death of mothers who attend secondary education was decreased by a factor of 0.351 (Φ = 0.351, 95% CI: 0.202, 0.608) as compared to mothers who had higher educational status by keeping the remaining covariates constant.

The survival time to death of children born to mothers who made health care decisions jointly with their husband/parent was shortened by a factor of 0.838 (Φ = 0.838, 95% CI: 0.623, 1.127) compared to children born to a mother who made their own health care decision (mothers alone) by keeping the remaining covariates constant. The survival time to death of children born to mothers who made health care decisions (husband/parent alone) and others was shortened by a factor of 0.49 (Φ = 0.49, 95% CI: 0.375, 0.664) compared to children born to mothers who made their own health care decisions (mothers alone) by keeping the remaining covariates constant.

The survival time to death of female children was increased by a factor of 2.149 (Φ = 2.149, 95% CI: 1.796, 2.570) as compared to male children by keeping the remaining covariates constant. The estimated acceleration factor of children among home-delivered mothers was estimated to be 0.618, indicating that the survival time to death of children among home-

**Table 7. AFT shared frailty model diagnostics and comparisons for time to death among under-five children and its predictors among live births in Africa.**

| Model Distribution | Frailty | Theta | AIC | BIC | LL | Deviance (−2LL) | LR test of theta With p-value |
|---|---|---|---|---|---|---|---|
| Exponential | Gamma | 0.0437894 | 151104.3 | 151486.5 | -75515.135 | 151,030.27 | 65.0 (p<0.001) |
| Exponential | Inverse Gaussian | 0.0455248 | 151103.3 | 151485.6 | -75514.662 | 151,029.324 | 65.95(p<0.001) |
| Weibull | Gamma | .0240196 | 114756.9 | 115149.4 | -57340.43 | 114,680.692 | 23.62 (p<0.001) |
| Weibull | Inverse Gaussian | .0243438 | 114756.7 | 115149.3 | -57340.346 | 114,680.692 | 23.79 (p<0.001) |
| Log-normal | Gamma | 0.0237737 | 113660.7 | 114053.2 | -56792.337 | 113,584.674 | 23.22(p<0.001) |

*Log normal inverse Gaussian and log-logistic gamma and* Inverse Gaussian *shared frailty model* didn't converge, NB: AIC = Akakian Information Criteria, BIC = Bayesian Information Criteria, LLR = Log-likelihood Ratio

delivered mothers was decreased by a factor of 0.618 ($\Phi$ = 0.618, 95% CI: 0.493, 0.773) than mothers delivered at a health facility by keeping the remaining covariates constant. The survival time to death of children born among mothers who have no ANC visits was shortened by a factor of 0.195 ($\Phi$ = 0.195, 95% CI: 0.146, 0.261) than mothers who have ≥4 ANC visits during pregnancy by keeping the remaining covariates constant. The survival time to death of children born among mothers who have 1–3 ANC visits was shortened by a factor of 0.732, ($\Phi$ = 0.195, 95% CI:0.93, 0.905) than mothers who have ≥4 ANC visits during pregnancy by keeping the remaining covariates constant. The survival time to death of children born among mothers who have 1–3 ANC visits was shortened by a factor of 0.732, ($\Phi$ = 0.195, 95% CI:0.93, 0.905) than mothers who have ≥4 ANC visits during pregnancy by keeping the remaining covariates constant. The survival time to death of children born among mothers who delivered by cesarean section was shortened by a factor of 0.228 ($\Phi$ = 0.228, 95% CI: (0.161, 0.323)) than mothers delivered by vaginal/spontaneous vaginal deliveries by keeping the remaining covariates constant. The acceleration factor and 95% confidence interval for mothers who have multiple births were 0.001, using singleton birth outcomes as a reference group. This suggests that mothers who have multiple birth outcomes exhibit decreased survival time for children as compared with the reference group (singleton birth outcome) ($\Phi$ = 0.001, 95% CI: 0.001, 0.0014). The acceleration factor and 95% confidence interval of children in the second, and third birth orders were 2.165 and 2.952, using the first birth order as a reference group. This suggests that children who had a second ($\Phi$ = 2.165, 95% CI: 1.582, 2.962), and third ($\Phi$ = 2.952, 95% CI: 2.096, 4.159) birth order exhibit increased survival time as compared with the reference group (first birth order). The acceleration factor and 95% confidence interval for children who were small in size at birth were 0.050, using average birth size at birth as a reference group. This suggests that children who were small in size at birth exhibit decreased survival time as compared with the reference group ($\Phi$ = 0.050, 95% CI: 0.038, 0.066). The acceleration factor and 95% confidence interval for children who were born in a community with low women's education were 0.725, using high community women's education as a reference group. This suggests that children who were born in a community with a low women's education exhibit decreased survival time as compared with the reference group ($\Phi$ = 0.725, 95% CI: 0.578, 0.909). The covariate wealth index was statistically determined for time to death. The acceleration factor and 95% confidence interval of having a poor wealth index for a group of wealth index were 0.754 (0.577, 0.985) when compared to the rich group (a reference category). This indicates children born from poor wealth index households or families have a shorter time to death than children born from rich families. The covariate respondents' working was statistically determined for the time to death. The acceleration factor and 95% confidence interval of women who had worked were 0.415 (0.337, 0.515) when compared to women who had no work (a reference category). This indicates children born from mothers who had work have a shorter time to death than children born from mothers who had no work.

The estimated coefficient of the parameter for maternal age at first birth, which was between 20–35 years was 0.430. The sign of the coefficient was positive, which implies an increase in the log of survival time, and hence, an elongated expected duration of time to death than maternal age at first birth between 8–9 (reference group). In another way, the acceleration factor and 95% confidence interval of mothers delivered at the age between 20 and 35 years, for their first birth were 1.538 (1.255, 1.885) times mothers delivered at the age between 8 and 9 years (reference group). This indicates mothers who delivered at the age between 20–35 years for their first birth had a longer time to die than whose mothers delivered at the age of 20–35 years for their first birth (reference group) at a 5% level of significance by keeping the other covariates constant (Table 8).

**Table 8. Multivariable lognormal AFT gamma shared frailty parametric survival regression analysis of time to death and its predictors among under-five children in Africa using recent DHS from 2014–2022.**

| Respondent's characteristic | Categories | coefficient | SE coefficient | Φ(TR) | SE Φ | 95% CI φ(TR) | P value |
|---|---|---|---|---|---|---|---|
| Place of residence | Urban | Ref | Ref | Ref | Ref | Ref | Ref |
| | Rural | -0.196 | 0.124 | 0.822 | 0.102 | (0.64,1.048) | 0.114 |
| Country | Eastern Africa | 1.049 | 0.310 | 2.855 | 0.885 | (1.555, 5.243) | 0.001 |
| | Western Africa | -0.705 | 0.311 | 0.494 | 0.154 | (0.268, 0.909) | 0.023 |
| | Northern Africa | 3.088 | 0.367 | 21.92 | 8.045 | (10.68, 45.01) | < 0.001* |
| | Central Africa | 0.297 | 0.318 | 1.345 | 0.428 | (0.721, 2.510) | 0.351 |
| | Southern Africa | Ref | Ref | Ref | Ref | Ref | Ref |
| Maternal age | 15–19 | 1.680 | 0.194 | 5.366 | 1.041 | (3.668, 7.848) | < 0.001* |
| | 20–35 | 0.129 | 0.238 | 1.137 | 0.270 | (0.714,1.811) | 0.588 |
| | 36–49 | Ref | Ref | Ref | Ref | Ref | Ref |
| Maternal Education status | No education | -1.249 | 0.296 | 0.286 | 0.085 | (0.16, 0.512) | < 0.001* |
| | Primary education | -1.536 | 0.289 | 0.215 | 0.062 | (0.122, 0.379) | < 0.001* |
| | 2ndary education | -1.047 | 0.281 | 0.351 | 0.098 | (0.202, 0.608) | < 0.001* |
| | Higher education | Ref | Ref | Ref | Ref | Ref | Ref |
| Maternal age at first birth | 8–19 | Ref | Ref | Ref | Ref | Ref | Ref |
| | 20–35 | 0.430 | 0.104 | 1.538 | 0.159 | (1.255, 1.885) | < 0.001* |
| | 36–49 | 0.211 | 0.812 | 1.231 | 1.003 | (0.251, 6.070) | 0.795 |
| wealth index combined | Poor | -0.282 | 0.136 | 0.754 | 0.103 | (0.577, 0.985) | 0.039 |
| | Middle | -0.245 | 0.141 | 0.782 | 0.11 | 0.594, 1.030) | 0.081 |
| | Rich | Ref | Ref | Ref | Ref | Ref | Ref |
| Sex of household head | Male | Ref | Ref | ref | Ref | Ref | Ref |
| | Female | -0.0904 | 0.115 | 0.914 | 0.105 | (0.729,1.144) | 0.432 |
| Respondents working | No | Ref | Ref | Ref | Ref | Ref | Ref |
| | Yes | -0.878 | 0.110 | 0.415 | 0.046 | (0.337, 0.515) | < 0.001* |
| Birth outcome | Single | Ref | Ref | ref | Ref | Ref | Ref |
| | Multiple | -7.045 | 0.227 | 0.001 | 0.0002 | (0.001, 0.0014) | < 0.001* |
| Birth order | First | Ref | Ref | ref | Ref | Ref | Ref |
| | Second | 0.772 | 0.160 | 2.165 | 0.346 | (1.582, 2.962) | < 0.001* |
| | Third | 1.083 | 0.175 | 2.952 | 0.51 | (2.096, 4.159) | < 0.001* |
| | 4+ | 0.232 | 0.161 | 1.262 | 0.203 | 0.921,1.729) | 0.148 |
| Place of delivery | Home | -0.482 | 0.115 | 0.618 | 0.07 | (0.493,0.773) | < 0.001* |
| | Health facility | Ref | Ref | ref | Ref | Ref | Ref |
| Mode of delivery | Vaginal | Ref | Ref | ref | Ref | Ref | Ref |
| | C/S | -1.478797 | 0.178 | 0.228 | 0.040 | (0.161, 0.323) | < 0.001* |
| Women's healthcare decision making autonomy | Respondent alone | Ref | Ref | Ref | Ref | Ref | Ref |
| | Jointly with their husband/parent | -0.177 | 0.151 | 0.838 | 0.126 | 0.623, 1.127) | 0.242 |
| | Husband/parent alone & others | -0.695 | 0.146 | 0.49 | 0.073 | (0.375, 0.664) | < 0.001* |
| Number of ANC visits | No visits | -1.632 | 0.148 | 0.195 | -11.06 | (0.146, 0.261) | < 0.001* |
| | 1–3 | -0.311 | 0.108 | 0.732 | -2.88 | (0.93, 0.905) | 0.004 |
| | ≥4 | Ref | Ref | Ref | Ref | Ref | Ref |
| Child size at birth | Small | -2.991 | 0.142 | 0.050 | 0.007 | (0.038, 0.066) | < 0.001* |
| | Average | Ref | Ref | Ref | Ref | Ref | Ref |
| | Large | -0.144 | 0.103 | 0.866 | 0.089 | 0.707,1.059) | 0.162 |
| Sex of child | Male | Ref | Ref | Ref | Ref | Ref | Ref |
| | Female | 0.765 | 0.091 | 2.149 | 0.196 | (1.796, 2.570) | < 0.001* |

(*Continued*)

**Table 8.** (Continued)

| Respondent's characteristic | Categories | coefficient | SE coefficient | Φ(TR) | SE Φ | 95% CI φ(TR) | P value |
|---|---|---|---|---|---|---|---|
| Household Media exposure | No | Ref | Ref | ref | Ref | Ref | Ref |
|  | Yes | -0.123 | 0.105 | 0.884 | 0.093 | 0.719, 1.087) | 0.242 |
| Community-women education | Low | -0.322 | 0.115 | 0.725 | 0.084 | 0.578, 0.909) | 0.005 |
|  | High | Ref | Ref | ref | Ref | Ref | Ref |
| Community poverty | Low | Ref | Ref | ref | Ref | Ref | Ref |
|  | High | 0.209 | 0.111 | 1.232 | 0.137 | (0.99,1.532) | 0.060 |
| Community ANC utilization | Low | 0.060 | 0.105 | 1.062 | 0.112 | (0.864,1.306) | 0.569 |
|  | High | Ref | Ref | Ref | Ref | Ref | Ref |

**Notes**: Likelihood ratio test of θ = 0: chi–square = 23.44 at P value < 0.001*; Statistically significant at P-value < 0.02; **statistically significant at P-value < 0.05; ***statistically significant at P-value < 0.001; 1 = reference

## 3.10. Measure of variation (random effect)

The Frailty model has an unobserved multiplicative effect on the hazard rate for all individuals in the same group. In the shared frailty model, under-five children in the same cluster share the same nuisance (frailty) factor. Parameter $\theta$ provides information on the variability (dependency) of the population in the same cluster. Under-five children in cluster I with $U_i > 1$ and $U_I < 1$ have a frailer than higher risk and lower risk, respectively. Based on the different frailty terms, one frailty term was employed using clusters taken as random effects. For a single frailty term, the model specification is given by [25]. $h_{ij}(t) = h_0(t)Z_i \cdot [\exp \beta x_{ij}]$, where $Z_i = \exp.(w_i)$ is the frailty for the ith country. $u_i$'s, i = 1, s, are the actual values of a sample from density $fU$. The parametric frailty model fitted using the Gompertz baseline hazard distributional assumption and the gamma frailty distribution model fitted in cluster taken as random effects frailty for the independent variables $Z_i$ is a random positive quantity with a mean of one (to identify the model) and variance $\theta$. Individuals with $Z_i > 1$ are said to be frailer for reasons that the covariates cannot explain, and they are at a higher risk of failure. Individuals with $Z_i < 1$ are less frail and have a higher chance of survival (assuming a specific covariate pattern). To identify potential risk factors for under-five mortality, a cluster-level parametric shared frailty survival model was applied. The shape parameter in the Gompertz baseline hazard distribution model was ($\rho = -0.256$; 95% CI: -0.262, -0.251). The negative shape parameter indicates that the risk of death among under-five children decreases exponentially as their age increases. The model estimated significant unobserved heterogeneity of under-five children death in the same cluster ($\theta = 0.06$, 95% CI: (0.0094, 0.022)). Kendall's tau ($\tau$) was used to calculate the correlation between two event times in the same country. Kendall's tau ($\tau$) determines the relationship between two events in the same country by dividing the frailty ($\theta$) by the two-plus frailty ($\theta$). Higher frailty ($\theta$) results in increased dependency and higher Kendall's tau ($\tau$) [26]. Kendall's tau ($\tau$) = ($\theta$) /($\theta$)+2, where ($\tau$)$\sum$ (0, 1). The dependence within the cluster was $\tau = 1.4\%$ (95% CI: 0.0094, 0.022), with the lowest dependency at 0.94% and the highest at 2.2% across the cluster.

## 4. Discussion

The overall goal of this study was to investigate the time to death of under-five mortality and its predictors in Africa using weighted nationally representative recent standard DHS data from 2014–2022 by using the Lognormal AFT gamma shared frailty model. According to this

study, Africa's overall under-five child death rate was 37.55 percent (95% confidence interval: 37.36 to 37.74) per 1000 live births. This study is in line with a study done in Bhutan (3.70%) [27]. However, it is higher than the average world index (3.4%) [28] and this study is lower than the study done in Somali 57/1000 live birth [29], Ethiopia (5.76% to 10.00%) [30–34]; Tanzania (7.04%) [35]; Ghana (4.91%) [36], Bengal (9.69%) [37], East Africa (5.132%) [38] and Sub-Saharan Africa (7.35%) [39]. The possible source of variation could be due to the country's differences in socioeconomic status, access to healthcare, regional variations in sociocultural and contextual factors, time variation, as well as differences in maternal education level.

By holding the other covariates constant, the survival time to death for children born in Western Africa was reduced by a factor of 0.494 ($\Phi$ = 0.494, 95% CI: (0.268, 0.909). The survival time to death of children born in Eastern Africa was increased by a factor of 2.855 ($\Phi$ = 2.855, 95% CI: 1.555, 5.243) by keeping the remaining covariates constant. It is essential to find a solution to the problem of regional differences in mortality among children under five. Numerous studies found a substantial relationship between child death rates under five and geographic location [22, 40–43]. These variations in under-five mortality rates around the region may be brought about by unequal access to medical facilities or by varying degrees of policies, programs, and interventions related to children's survival. Investigating the causes and creating intervention plans are essential to closing the disparity.

The current study found that, when all other covariates were held constant, the estimated acceleration factor of children born to mothers between the ages of 15 and 19 was 5.366. This means that, compared to children born to mothers between the ages of 36 and 49, the survival time to death of children born to mothers aged 15 to 19 was increased by 5.366 ($\Phi$ = 5.366, 95% CI: 3.668, 7.848). This finding was supported by previous studies conducted in Kenya [44], Uganda [45], and Afghanistan [46]; Nigeria [47], East Africa [38], and sub-Saharan Africa [18, 48]. Adolescent mothers are more likely to have children with precarious health outcomes and a higher risk of under-five mortality [49]. Several studies [42], have demonstrated that the younger age of women during childbirth (often less than 20 years old) exhibited noticeably increased odds of under-five mortality compared to middle-aged [17, 40, 41, 47, 50, 51] during childbearing. One plausible explanation is that early-life dysfunctional biological and social mechanisms negatively impact the health of the mother's firstborn child. Mature mothers who give birth to children run the risk of malnourishment, low birth weight, and congenital conditions, including Down syndrome [18].

The study's estimated acceleration factor for mothers without formal education was found to be 0.286. This means that, when all other covariates are held constant, mothers without formal education had a 0.286 shorter survival time to death ($\Phi$ = 0.286, 95% CI: (0.16, 0.512) compared to mothers with higher educational status. The estimated acceleration factor for mothers who had primary education was estimated to be 0.215, indicating that the survival time to death of mothers who had primary education was decreased by a factor of 0.215 ($\Phi$ = 0.215, 95% CI: 0.122, 0.379) than mothers who had higher educational status by keeping the remaining covariates constant. The estimated acceleration factor for mothers who attend secondary education was estimated to be 0.351, indicating that the survival time to death of mothers who attend secondary education was decreased by a factor of 0.351 ($\Phi$ = 0.351, 95% CI: 0.202, 0.608) compared to mothers who had higher educational status by keeping the remaining covariates constant. Reducing mortality among children under five seemed to be significantly influenced by mothers' education [18, 17, 50–56]. Compared to their counterparts, an educated woman appears to be more wise and aware of the value of using healthcare, nutrition, and cleanliness to improve the health of their child [57, 58]. Also, the current study was consistent with studies conducted in SSA [18, 59], East Africa [38], India [60], Nigeria [61], and Ghana [36]. This could be because mothers with higher levels of education are more conscious

of the value of immunizing children and using healthcare services. This may lower the chance of fatalities among children [38].

The survival time to death of female children was increased by a factor of 2.149 ($\Phi$ = 2.149, 95% CI: 1.796, 2.570) than male children by keeping the remaining covariates constant. The finding was in agreement with studies conducted in Ethiopia [30–33, 62–65], Malawi [66], Ghana [36], Nigeria [17, 67]; Sierra Leone [68]; Tanzania [35]; Uganda, [69], East Africa [38], Sub-Saharan Africa [39]; India [70], Iraq [71]. This could be due to biological variations such as circulatory hormones and innate defense mechanisms for various illnesses. Accordingly, it suggests that because of their fatal prognosis, male babies require extra care while they adjust to their new surroundings [38].

Keeping the other covariates constant, the estimated acceleration factor of children among mothers delivered at home was estimated to be 0.618, meaning that the survival time to death of children among mothers delivered at home was decreased by a factor of 0.618 ($\Phi$ = 0.618, 95% CI: 0.493, 0.773) compared to mothers delivered at a health facility. This finding was supported by studies reported in Nigeria [17], SSA [18, 39], and East Africa [38]. One plausible explanation could be that health facility delivery acts as the primary entry point for the utilization of essential pediatric healthcare services, such as PNC visits and childhood vaccinations, which could potentially lower the rate of under-five mortality from diseases like malaria, pneumonia, and diarrhea [38].

The survival time to death of children born among mothers who have no ANC visits was shortened by a factor of 0.195 ($\Phi$ = 0.195, 95% CI: 0.146, 0.261) than mothers who have $\geq$4 ANC visits during pregnancy by keeping the remaining covariates constant. The survival time to death of children born among mothers who have 1–3 ANC visits was shortened by a factor of 0.732, ($\Phi$ = 0.195, 95% CI:0.93, 0.905) than mothers who have $\geq$4 ANC visits during pregnancy by keeping the remaining covariates constant. Results from earlier studies revealed that an increase in the number of ANC visits reduces under-five mortality [22, 42, 50, 51]. This finding suggests implementing appropriate intervention programs to encourage ANC visits to significantly reduce under-five mortality. The survival time to death of children born among mothers who delivered by cesarean section was shortened by a factor of 0.228 ($\Phi$ = 0.228, 95% CI: (0.161, 0.323)) than mothers delivered by vaginal or spontaneous vaginal deliveries by keeping the remaining covariates constant. This finding was supported by the findings of studies in Sub-Saharan Africa [18]. The explanation may be the higher risk of preterm birth, birth abnormalities, neonatal infection, lack of access to quality treatment, and maternal complications that could result in under-five mortality, all of which are linked to cesarean sections [72, 73].

When comparing the acceleration factor for mothers of multiple births to the reference group of singleton birth outcomes, the results were 0.001. This implies that, compared to the reference group (singleton birth result), mothers with multiple birth outcomes have children with shorter survival times ($\Pi$ = 0.001, 95% CI: 0.001, 0.0014). This finding is supported by previous studies conducted in the United States [74], Ethiopia [33], and Sub-Saharan Africa [18]. Different studies have revealed that multiple births have also affected the health of under-five children [18, 40, 42, 50, 75, 76]. The possible justification might be biological factors [77]. Besides, multiple births increase individual family size, which leads to prenatal attention per child diminishing [78]. Several common causes include inadequate handling of multiple births, increased likelihood of birth malformations in multiple births, increased pregnancy risk compared to single birth, potential for growth retardation or premature birth, and additional delivery problems [62, 79].

Using the first birth order as a reference group, the acceleration factor and 95% confidence interval for the second and third birth order infants were 2.165 and 2.952, respectively. In

comparison to the reference group (first birth order), it appears that children with second ($\Pi$ = 2.165, 95% CI: 1.582, 2.962) and third ($\Phi$ = 2.952, 95% CI: 2.096, 4.159) birth orders have longer lifetimes. Additionally, compared to children in the middle, first-born children and those born with an order of four or higher show a higher risk of mortality in various studies that have been conducted (albeit these studies vary from one another) [50, 79]. This result was in line with the study's findings in Nigeria [47]; East Africa [38], and India [80]. The most likely explanation is that the first child is more likely to die before turning five because of pregnancy-related issues such as preeclampsia, fetal distress, antepartum hemorrhage, and preterm [38].

Using the average birth weight as a reference group, the acceleration factor and 95% confidence interval for infants who were small at birth were found to be 0.050 in this study. This indicates that, compared to the reference group, children who were smaller at birth had shorter survival times ($\Pi$ = 0.050, 95% CI: 0.038, 0.066). This result is consistent with the studies done in Nigeria [81], Ghana [53], Tanzania [54], Ethiopia [30, 31, 82, 83]; Malawi [66], Tanzania [54], and Nigeria [17, 67]; Sierra Leone [68], East Africa [38]; Sub-Saharan Africa [39]. On the other hand, during delivery, exceptionally large newborns had distinct birth traumas. Additionally, small-sized babies are more likely to be born prematurely, to contract infections, and to have birth abnormalities and problems, all of which increase the risk of death for children under five [38].

The covariate wealth index was statistically determined for time to death. The acceleration factor and 95% confidence interval of having a poor wealth index for a group of wealth index were 0.754 (0.577, 0.985) when compared to the rich group (as a reference category). This finding was supported by the findings of Bangladesh [84], Ghana [85], East Africa [38], and Sub-Saharan Africa. The possible explanation could be that children from wealthier households might have better child healthcare services and good nutrition, including appropriate breastfeeding, sunlight exposure, and immunization [39]. This indicates children born from poor wealth index households or families have a shorter time to death than children born from rich families. Poor families are compelled to have higher risks of under-five mortality compared to rich families [31, 39]. There are a few possible causes for this discrepancy. Poor families find it difficult to pay for necessary but expensive health care services; they may also fail to give mothers enough nourishing food and may need to be more informed about the general health care of both mother and child [18, 56].

For the period to death, the covariate respondents' working was statistically determined. Comparing women with no work (as the reference category) to those who had, the acceleration factor and 95% confidence interval for the former group were 0.415 (0.337, 0.515). This suggests that children born to working mothers die younger than children born to unemployed mothers. Many studies have shown a higher risk of under-five mortality in those mothers who are employed [41, 55, 56, 86]. This kind of finding suggests that rather than preventing working women from working, there should be workable options for child care.

## 5. Conclusion

The under-five mortality rate in Africa is still too high, despite the government and other health partners making efforts to enhance child survival. The most significant factors influencing under-five child mortality in Africa, as indicated by this study, are country region, maternal age, maternal education status, maternal age at first birth, respondents working, birth outcome, wealth index combined, birth order, place of delivery, mode of delivery, women's health care decision-making autonomy, number of ANC visits, child size at birth, sex of child, and community women's education. This research was carried out employing recent data

from the Demographic Health Survey (DHS) from 2014 to 2022. It is imperative to address the prominent risk factors that contribute to high rates of childhood mortality by developing and implementing timely and effective interventions. The direct reduction of the main causes of mortality among children under five should be the goal of these initiatives. Public health initiatives can significantly reduce childhood mortality rates worldwide by strategically and empirically addressing the main causes and risk factors. To address the intricate network of social, economic, environmental, and healthcare-related issues that lead to high under-five mortality, a thorough, multidimensional strategy is required. Significant progress may be made in preserving the lives of young children and maintaining their right to a good start in life through coordinated, well-planned actions.

## Supporting information

**S1 Table. Life table showing the survival to death among under-5 children in Africa using recent DHS from 2014–2022 (n = 226862).**
(DOCX)

**S2 Table. Cumulative hazard function among children in Africa using recent DHS 2023 (n = 226862).**
(DOCX)

**S1 File. Stata data used for analysis.**
(DTA)

## Author Contributions

**Conceptualization:** Bikis Liyew.

**Data curation:** Bikis Liyew.

**Formal analysis:** Bikis Liyew, Kemal Tesfa, Kassaye Demeke Altaye, Abeje Diress Gelaw, Alemu Teshale Bicha, Ayanaw Guade Mamo, Kassaw Chekole Adane.

**Investigation:** Bikis Liyew, Kassaw Chekole Adane.

**Methodology:** Bikis Liyew, Kemal Tesfa, Kassaye Demeke Altaye, Abeje Diress Gelaw, Alemu Teshale Bicha, Ayanaw Guade Mamo.

**Software:** Kemal Tesfa, Kassaye Demeke Altaye, Abeje Diress Gelaw, Alemu Teshale Bicha, Ayanaw Guade Mamo, Kassaw Chekole Adane.

**Supervision:** Kemal Tesfa, Ayanaw Guade Mamo, Kassaw Chekole Adane.

**Validation:** Kemal Tesfa, Kassaye Demeke Altaye, Alemu Teshale Bicha, Ayanaw Guade Mamo, Kassaw Chekole Adane.

**Visualization:** Kassaw Chekole Adane.

**Writing – original draft:** Bikis Liyew, Kassaye Demeke Altaye, Alemu Teshale Bicha, Kassaw Chekole Adane.

**Writing – review & editing:** Bikis Liyew, Kemal Tesfa, Abeje Diress Gelaw, Ayanaw Guade Mamo, Kassaw Chekole Adane.

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
