## [Decision Letter · Decision Letter 0]

19 Sep 2024

PONE-D-24-23176Modeling Survival Time to Death among Under-Five Children in Africa: Application of Lognormal AFT Gamma Shared Frailty Parametric Survival Regression ModelPLOS ONE

Dear Dr. Liyew,

Thank you for submitting your manuscript to PLOS ONE. After careful consideration, we feel that it has merit but does not fully meet PLOS ONE’s publication criteria as it currently stands. Therefore, we invite you to submit a revised version of the manuscript that addresses the points raised during the review process.

The manuscript has been evaluated by two reviewers, and their comments are available below.

The reviewers have raised a number of major concerns. They feel the manuscript should outline a clearly-defined research question, and they request improvements to the reporting of methodological aspects of the study and result, as well as improved framing in the introduction. 

Could you please carefully revise the manuscript to address all comments raised?

We look forward to receiving your revised manuscript.

Kind regards,

Avanti Dey, PHD

Staff Editor

PLOS ONE

Journal Requirements:

Additional Editor Comments (if provided):

Reviewers' comments:

Reviewer's Responses to Questions

**Comments to the Author**

1. Is the manuscript technically sound, and do the data support the conclusions?

Reviewer #1: Partly

Reviewer #2: Partly

2. Has the statistical analysis been performed appropriately and rigorously? 

Reviewer #1: Yes

Reviewer #2: Yes

3. Have the authors made all data underlying the findings in their manuscript fully available?

Reviewer #1: Yes

Reviewer #2: No

4. Is the manuscript presented in an intelligible fashion and written in standard English?

Reviewer #1: No

Reviewer #2: Yes

5. Review Comments to the Author

Reviewer #1: The authors seem to have rushed to submit the manuscript prematurely.

In its present form, the manus is too scattered and disorganized with

Of the 44 - 2 = 42 pages from Introduction to Conclusion 24 pages (57 %) are or include tables. More seriously, these 24 pages appear in the body of the manuscript, not on Appendix which could allow for better reading of the text.

To the surprise of this reviewer, the authors have pasted the life-table (or Kaplan-Meier) estimates of the survivor function on 4 pages (pages 17-20) and the corresponding estimates of the hazard function on another 4 pages (page 21 - 24). This indicates they are well versed with how results are presented in scientific articles. Seasoned researchers could have chosen to present summary statistics of the functions in a table and possibly plot the surivivor and hazard functions on one page - thereby reducing their 8 pages to 1-2 pages.

The manuscript seems to have 4-5 Sections:

- Introduction on pages 3-4 (which should be 1-2),

- Methods and Materials on pages 5-8 (should be 3-6)

- Results on pages 9-39!!! (should be 7-37)

- Discussion on pages 39 - 43 (should be 37-41)

- Conclusion on page 44 (should be 42)

There also seems to be many sub-sections within each section. Unfortunately, it is not possible to distinguish which is a Section and which is a sub-section because they are neither numbered not can be identified by their font-sizes and/or font-types.

The above make it hard or impossible to extract the merits (if any) of the manuscript. The authors, therefore, need to, at least, do major revision to polish their manuscript if they believe they have something worth publishing in journals like PLOS ONE. Below are some recommendations:

1. They can begin with the title. First it's not appropriate to use AFT in the title (though the reviewer knows it means Accelerated Failure Time). Further, the title can be modified (shortened) depending on whether the focus is on methodology or the substantive issues. For instance, "Parametric Modeling of Under-5 Survival Among 30 African Countries" or "Under-5 Survival in 30 African Countries: The Use of Parametric Models, etc... can be some alternatives.

2. They also need to justify the choice of log-normal model for the survival time and gamma model for the frailty. They also need to explain why they need failty models (what new insights do they provide compared to standard models).

3. They need to write their manuscript concisely and coherently with well organized and clear sections and subsections.

4. Tables and plots of relevant results should be put in an Appendix (at least at the stage of review).

5. Make it clear right at the outset the expected contribution of the study, for instance, is it a methodological contribution (to show the merits of parametric modeling), or a substantive contribution to report the status of under-5 mortality in the 30 countries (or both)?

There is much more to be done but I hope the above will suffice as a guideline.

Reviewer #2: Date: 14-sep-2024

General Comments

Thank you for inviting me to review a manuscript entitled “Modeling Survival Time to Death among Under-Five Children in Africa: Application of Lognormal AFT Gamma Shared Frailty Parametric Survival Regression Model”. This is an interesting study, and the authors have collected a good dataset using the critical advantage methodology. The paper is generally well written and structured. However, from my point of view, the manuscript has some shortcoming with respect to some dataset analyses and text, and I feel this good dataset has not been used to its full extent. Underneath, the reviewer has provided several remarks on the text, as it is often vague and long-winded. In several examples, I also recommended citing more relevant and recent literature. Additionally, the reviewer has made additional comments for more in-depth analyses of the dataset. I have assessed the manuscript, and I can hardly suggest it for publication for the following reasons:

Key critical points are:

a) Please check the grammar carefully in the overall manuscript; somewhere it’s written in the past and somewhere in the present and somewhere it needs punctuations. Check the spelling mistakes; there are major spelling errors in the overall document. If possible, please I suggest that this manuscript be reviewed by an English-language expert.

Specific Comments

1. The Background section on the body of manuscript: this section needs revision so that it will cover the nature of the core problem, the known aspect of problem (i.e., what is known about the core problem), the unknown aspect of the problem (What is unknown about the core problem), and the expectations from the current study (what and how the current study will add to the existing knowledge).

Abstract

Background

2. Is that true “Despite the significant progress made in reducing under-five mortality rates across African nations, the mortality rate in this age group continues to be the highest globally.” What does it mean? Please write again?

Method

3. “This study employed a multivariable lognormal accelerated failure time gamma shared frailty parametric survival regression analysis to identify the predictors of time-to-death among infants in these African nations.” Is this study for infants? Or for under five children?

Conclusion

4. In this study country region, maternal age, maternal education status, maternal age at first birth, respondent's employment status, birth outcome, wealth index, birth order, place of delivery, mode of delivery, women's autonomy in healthcare decision-making, number of antenatal care visits, child's size at birth, sex of the neonate, and community-level women's education. So, what? Please again and again and make a sense?

Method

5. The variables used in this study were extracted, cleaned, and processed using STATA version 14 software. Please remove it? This is repetition?

6. When we prepare manuscripts in Plos One no need of “Keywords: survival analysis, accelerated failure time model, Africa.” Please read guide line for Plos One?

7. Please remove the citation style of (1), (2) of endnote style instead of this [1], [2]

8. Please use the latest reference rather than old reference? Like reference 2, 7, 9, 10, 49, 50, 51, 74 …etc…

9. How do you merge these 30 country datasets please give a clear justification?

10. Please include the STATA (version 14) commands that used for final model analysis as appendix (supplementary materials)?

6. PLOS authors have the option to publish the peer review history of their article (what does this mean?). If published, this will include your full peer review and any attached files.

Reviewer #1: No

Reviewer #2: **Yes: **Gebru Gebremeskel Gebrerufael

---

## [Author Response · Author response to Decision Letter 0]

16 Oct 2024

Authors’ Response for Reviewers’ Comments

Manuscript ID: PONE-D-24-23176

Title: Parametric Modeling of Under-5 Children Survival Among 30 African Countries: Lognormal Accelerated Failure Time Gamma Shared Frailty Model

Dear editor(s) and reviewers

First for all the authors would like to thank the editor(s) and reviewers for your precious time, thoughtful comments and constructive suggestions, which help to improve the quality of this manuscript. We have responded to each critique/ comment and believe that the manuscript is much improved with the changes we made as suggested by the editor and reviewers. The corresponding changes and refinements made in the revised manuscript are summarized in our response below.

Response=> Authors response for editor and/ reviewers’ comments 

Reviewer #1

The authors seem to have rushed to submit the manuscript prematurely.

Response: We thank the reviewer for raising this important point. We appreciate your concern regarding the manuscript's readiness. To be honest, we took the necessary time to thoroughly review and revise the content to ensure the quality of manuscript. Therefore, your insights are valuable in helping us enhance the quality of our work.

In its present form, the manus is too scattered and disorganized with 

Response: Thank you for your constructive feedback. We made a comprehensive revision to improve the structure and coherence of the content, ensuring that key points are presented clearly and logically. Your feedback guides us to re organize the manuscript headings, sub-headings and also other typo errors to enhancing the overall clarity and flow of the manuscript.

Of the 44 - 2 = 42 pages from Introduction to Conclusion 24 pages (57 %) are or include tables. More seriously, these 24 pages appear in the body of the manuscript, not on Appendix which could allow for better reading of the text.

Response: Thank you for your insightful comment. We recognize that the inclusion of many tables within the main body of the manuscript may disrupt the reading flow. To address the issue, we moved some tables to supplementary files, ensuring that the text remains more cohesive and accessible. We appreciate your suggestion and we made corrections and highlighted in red it in the revised manuscript with track change.

To the surprise of this reviewer, the authors have pasted the life-table (or Kaplan-Meier) estimates of the survivor function on 4 pages (pages 17-20) and the corresponding estimates of the hazard function on another 4 pages (page 21 - 24). This indicates they are well versed with how results are presented in scientific articles. Seasoned researchers could have chosen to present summary statistics of the functions in a table and possibly plot the survivor and hazard functions on one page - thereby reducing their 8 pages to 1-2 pages.

Response: thank you for your suggestion. The suggested correction has been made and highlighted in track change.

The manuscript seems to have 4-5 Sections: Introduction on pages 3-4 (which should be 1-2), Methods and Materials on pages 5-8 (should be 3-6); Results on pages 9-39!!! (should be 7-37), Discussion on pages 39 - 43 (should be 37-41), Conclusion on page 44 (should be 42)

Response: Thank you for your detailed feedback. We appreciate your observations regarding the organization of the sections. We revised the manuscript to align the sections as you suggested, numbers for the Methods, Results, Discussion, and Conclusion, highlighted in mothers document.

There also seems to be many sub-sections within each section. Unfortunately, it is not possible to distinguish which is a Section and which is a sub-section because they are neither numbered not can be identified by their font-sizes and/or font-types.

Response: thank you for your suggestion. The suggested correction has been made and highlighted in track change.

The above make it hard or impossible to extract the merits (if any) of the manuscript. The authors, therefore, need to, at least, do major revision to polish their manuscript if they believe they have something worth publishing in journals like PLOS ONE. Below are some recommendations:

Response: Thank you, the reviewer, for the comments

1. They can begin with the title. First, it's not appropriate to use AFT in the title (though the reviewer knows it means Accelerated Failure Time). Further, the title can be modified (shortened) depending on whether the focus is on methodology or the substantive issues. For instance, "Parametric Modeling of Under-5 Survival Among 30 African Countries" or "Under-5 Survival in 30 African Countries: The Use of Parametric Models, etc... can be some alternatives.

Response: With great thanks the suggested correction has been made and highlighted in the track change

2. They also need to justify the choice of the log-normal model for the survival time and the gamma model for the frailty. They also need to explain why they need frailty models (what new insights do they provide compared to standard models).

Response: We thank the reviewer for raising this important point. In this survival analysis, we follow all the procedures based on the data-driven model of analysis. First, we test the proportional hazard assumption for Cox regression. However, the global test is significant, the PH assumption is violated. Due to the PH assumption violation, we go to parametric survival analysis to know the effect of cluster variation in survival analysis mixed effect models both random (a measure of variation) and fixed effect (a measure of association) were used. To account for this in survival analysis frailty model to account for unobserved heterogeneity was used. Regarding PH and AFT models: the PH model assumes hazard constant over time but may not be true at all times. It applies the comparison of hazard time. The effects of covariates or IDV is multiplicative with respect to hazard. While AFT applies a comparison of survival time. The effect of independent variables or covariates is multiplicative with respect to hazard. When the proportional hazards (PH) assumption underlying the PH approach is not met, the accelerated failure time (AFT) approach offers an alternative for analyzing time-to-event data, making it appropriate even when hazards are not proportional. Notably, this family of models includes a specific case that adheres to the PH assumption. AFT models focus on the time until an event occurs and can effectively handle data where hazards are not proportional.

Regarding log-normal model for the survival time and the gamma model for the frailty: in survival analysis the time variable is positively skewed data, which is not normally distributed, based on this the semiparametric model cox regression without the assumption of distribution is used. However, in our analysis we used parametric survival regression models which assumes different distribution based on the baseline AFT model like exponential AFT model, weibull AFT model, log normal AFT model and log-logistic AFT model. For frailty we have two common distributions: gamma and inverse gaussian distribution were used to account for unobserved heterogeneity. Finaly, we compare all these models based on AIC, BIC, LLH, and deviance value. Based on this information, the final fitted analysis model, the lognormal AFT gamma shared frailty model, was found to be the best fit of the five parametric models, which have the lowest AIC. 

3. They need to write their manuscript concisely and coherently with well-organized and clear sections and subsections.

Response: The suggested correction has been made and highlighted in the revised track change document.

4. Tables and plots of relevant results should be put in an Appendix (at least at the review stage).

Response: We recognize that including many tables within the main body of the manuscript may disrupt the reading flow. To address the issue, we moved some tables to supplementary files, ensuring that the text remains more cohesive and accessible.

5. Make it clear right at the outset the expected contribution of the study, for instance, is it a methodological contribution (to show the merits of parametric modeling), or a substantive contribution to report the status of under-5 mortality in the 30 countries (or both)?

Response: Thank you for the comment. More detailed information has been provided on these aspects in the introduction section. The persistent and substantial disparities in under-five mortality rates across different regions within Africa continue to present a significant, widespread challenge for all countries on the continent. Previous studies failed to account for the survival nature of under-five mortality and neglected the clustering effects in their data, creating a methodological gap. This study addresses these issues by employing a two-level frailty survival analysis to accurately control for cluster effects. 

There is much more to be done but I hope the above will suffice as a guideline.

Response: We very much appreciate this helpful comment. We are grateful for this comment as it points to an important point of view. This comment is crucial for this manuscript which was our gap. All type errors, sentence structure, factor interpretation, and language usage were resolved and copyedited to the revised document.

Reviewer #2

General Comments

Thank you for inviting me to review a manuscript entitled “Modeling Survival Time to Death among Under-Five Children in Africa: Application of Lognormal AFT Gamma Shared Frailty Parametric Survival Regression Model”. This is an interesting study, and the authors have collected a good dataset using the critical advantage methodology. The paper is generally well written and structured. However, from my point of view, the manuscript has some shortcoming with respect to some dataset analyses and text, and I feel this good dataset has not been used to its full extent. Underneath, the reviewer has provided several remarks on the text, as it is often vague and long-winded. In several examples, I also recommended citing more relevant and recent literature. Additionally, the reviewer has made additional comments for more in-depth analyses of the dataset. I have assessed the manuscript, and I can hardly suggest it for publication for the following reasons:

Response: We appreciate the positive feedback from the reviewer. All The suggested correction and comments has been made and highlighted in track change in whole mother document. 

Key critical points are:

a) Please check the grammar carefully in the overall manuscript; somewhere it’s written in the past and somewhere in the present and somewhere it needs punctuations. Check the spelling mistakes; there are major spelling errors in the overall document. If possible, please I suggest that this manuscript be reviewed by an English-language expert.

Response: We thank the reviewer for the suggestions. we have revised again the document. We have also noted repetitive sentences, unnecessary capitals and smalls, contextual spelling and sentence structure problems. Then we have re-arranged and re-edited the vague sentences. Hence clarity problems are resolved and highlighted by track change in the whole revised manuscript

Specific Comments

1. The Background section on the body of manuscript: this section needs revision so that it will cover the nature of the core problem, the known aspect of problem (i.e., what is known about the core problem), the unknown aspect of the problem (What is unknown about the core problem), and the expectations from the current study (what and how the current study will add to the existing knowledge).

Response: We thank the reviewer for raising this important point. We rewrote the introduction for greater focus. This section has been edited for conciseness by presenting the crucial points of what is known and not known, and why we carried out the study.

Abstract

Background

2. Is that true “Despite the significant progress made in reducing under-five mortality rates across African nations, the mortality rate in this age group continues to be the highest globally.” What does it mean? Please write again?

Response: Thank you for your view, The suggested correction has been made and highlighted in track change.

Method

3. “This study employed a multivariable lognormal accelerated failure time gamma shared frailty parametric survival regression analysis to identify the predictors of time-to-death among infants in these African nations.” Is this study for infants? Or for under five children?

Response: sorry for the typological error that we have made. The correction has been made and highlighted in track change. 

Conclusion

4. In this study country region, maternal age, maternal education status, maternal age at first birth, respondent's employment status, birth outcome, wealth index, birth order, place of delivery, mode of delivery, women's autonomy in healthcare decision-making, number of antenatal care visits, child's size at birth, sex of the neonate, and community-level women's education. So, what? Please again and again and make a sense?

Response: Thank you for your view. With great thanks the suggested correction has been made and highlighted in track change.

Method

5. The variables used in this study were extracted, cleaned, and processed using STATA version 14 software. Please remove it? This is repetition?

Response: With great thanks the suggested correction has been made and highlighted in track change

6. When we prepare manuscripts in Plos One no need of “Keywords: survival analysis, accelerated failure time model, Africa.” Please read guide line for Plos One?

Response: thank you for your suggestion. The suggested correction has been made and highlighted in track change.

7. Please remove the citation style of (1), (2) of endnote style instead of this [1], [2]

Response: thank you for your suggestion. The suggested correction has been made

8. Please use the latest reference rather than old reference? Like reference 2, 7, 9, 10, 49, 50, 51, 74 …etc…

Response: thank you for your suggestion. The suggested correction has been made

9. How do you merge these 30 country datasets please give a clear justification?

Response: first we start to think Africa as a continent how much the burden of under five mortality and survival status. Based on this as we know all countries have standard DHS data. We took each countries recent DHS dataset starting from 2014. Based on our criteria (recent standard DHS dataset) we took 30 countries. After this DHS data variable coding is consistent and similar across all countries. Therefore, we use two concepts to manage the data, is that data merging and appending. Append (combine observation): combine multiple datasets with similar variable coding to add different observations. Merge (combine variables): combine multiple data sets with different variables. All 30 countries data were saved in the form of stata file(.dta). all the datasets have common variable or ID to append, which is cluster number (V001). Based on this, we use stata software to append (data-------combine datasets----append datasets). First, we maximize stata software memory by using command set maxvar 32000. Then after by using cluster number we append all 30 countries and saved as a one file. 

10. Please include the STATA (version 14) commands that used for final model analysis as an appendix (supplementary materials)?

Response: The suggested correction has been mad. We added final model stata command as supplementary information. 

Generally, we have consumed more time and energy for re-edition of the vague sentences. Based on this, we have rewritten the whole document in a more understandable manner to resolve language problems. Furthermore, we have modified, added, and changed a lot of things start from the title up to references based on other reviewers in addition to you. We are lucky, since the manuscript is assigned for two peer reviewers, and all of the three reviewers’ comments and questions are different which helps us to learn a lot and modify the whole document.

---

## [Decision Letter · Decision Letter 1]

23 Dec 2024

Parametric Modeling of Under-5 Children Survival Among 30 African Countries: Lognormal Accelerated Failure Time Gamma Shared Frailty Model

PONE-D-24-23176R1

Dear Dr. Liyew,

We’re pleased to inform you that your manuscript has been judged scientifically suitable for publication and will be formally accepted for publication once it meets all outstanding technical requirements.

Kind regards,

James Mockridge

Staff Editor

PLOS ONE

Additional Editor Comments (optional):

Reviewers' comments:

Reviewer's Responses to Questions

**Comments to the Author**

1. If the authors have adequately addressed your comments raised in a previous round of review and you feel that this manuscript is now acceptable for publication, you may indicate that here to bypass the “Comments to the Author” section, enter your conflict of interest statement in the “Confidential to Editor” section, and submit your "Accept" recommendation.

Reviewer #2: All comments have been addressed

Reviewer #3: All comments have been addressed

2. Is the manuscript technically sound, and do the data support the conclusions?

Reviewer #2: Yes

Reviewer #3: (No Response)

3. Has the statistical analysis been performed appropriately and rigorously? 

Reviewer #2: Yes

Reviewer #3: (No Response)

4. Have the authors made all data underlying the findings in their manuscript fully available?

Reviewer #2: Yes

Reviewer #3: (No Response)

5. Is the manuscript presented in an intelligible fashion and written in standard English?

Reviewer #2: Yes

Reviewer #3: (No Response)

6. Review Comments to the Author

Reviewer #2: Title: “Parametric Modeling of Under-5 Children Survival Among 30 African Countries: Lognormal Accelerated Failure Time Gamma Shared Frailty Model”

Version: 2 Date: 24-Sep-24

Thank you for the opportunity to review the manuscript entitled ““Parametric Modeling of Under-5 Children Survival Among 30 African Countries: Lognormal Accelerated Failure Time Gamma Shared Frailty Model””.

The authors describe a secondary analysis of the dhs dataset to determine the “prevalence and predictors of Under-5 Children among 30 African Countries using Lognormal Accelerated Failure Time Gamma Shared Frailty Model”. This is the second review of this paper following revisions.

∼ The authors have successfully addressed all comments and have vastly improved the article.

I recommend this move forward for publication.

Reviewer #3: (No Response)

7. PLOS authors have the option to publish the peer review history of their article (what does this mean?). If published, this will include your full peer review and any attached files.

Reviewer #2: **Yes: **Gebru Gebremeskel Gebrerufael

Reviewer #3: No

---

## [Editor Report · Acceptance letter]

29 Dec 2024

PONE-D-24-23176R1 

PLOS ONE

Dear Dr. Liyew, 

I'm pleased to inform you that your manuscript has been deemed suitable for publication in PLOS ONE. Congratulations! Your manuscript is now being handed over to our production team.

Kind regards, 

on behalf of

Dr James Mockridge 

Staff Editor

PLOS ONE